# Heterophilic and homophilic cadherin interactions in intestinal intermicrovillar links are species dependent

**Michelle E. Gray**[1,2‡], **Zachary R. Johnson**[2‡¤], **Debadrita Modak**[2‡],
**Elakkiya Tamilselvan**[2,3], **Matthew J. Tyska**[4], **Marcos Sotomayor**[1,2,3]*

**1** Ohio State Biochemistry Program, The Ohio State University, Columbus, Ohio, United States of America,
**2** Department of Chemistry and Biochemistry, The Ohio State University, Columbus, Ohio, United States of
America, **3** Biophysics Program, The Ohio State University, Columbus, Ohio, United States of America,
**4** Department of Cell and Developmental Biology, Vanderbilt University School of Medicine, Nashville,
Tennessee, United States of America

¤ Current address: Marshall University, Joan C. Edwards School of Medicine, Huntington, West Virginia,
United States of America
‡ These authors share first authorship on this work.
* sotomayor.8@osu.edu

doi.org/10.1371/journal.pbio.3001463

Institute, UNITED STATES

**Data Availability Statement:** All relevant data are
within the paper and its Supporting information
files, with additional structural files deposited in the
protein data bank (accession codes 5CZR, 5CYZ,

## Abstract

Enterocytes are specialized epithelial cells lining the luminal surface of the small intestine
that build densely packed arrays of microvilli known as brush borders. These microvilli drive
nutrient absorption and are arranged in a hexagonal pattern maintained by intermicrovillar
links formed by 2 nonclassical members of the cadherin superfamily of calcium-dependent
cell adhesion proteins: protocadherin-24 (PCDH24, also known as CDHR2) and the mucin-
like protocadherin (CDHR5). The extracellular domains of these proteins are involved in het-
erophilic and homophilic interactions important for intermicrovillar function, yet the structural
determinants of these interactions remain unresolved. Here, we present X-ray crystal struc-
tures of the PCDH24 and CDHR5 extracellular tips and analyze their species-specific fea-
tures relevant for adhesive interactions. In parallel, we use binding assays to identify the
PCDH24 and CDHR5 domains involved in both heterophilic and homophilic adhesion for
human and mouse proteins. Our results suggest that homophilic and heterophilic interac-
tions involving PCDH24 and CDHR5 are species dependent with unique and distinct mini-
mal adhesive units.

## Introduction

Enterocytes are specialized epithelial cells lining the luminal surface of the small intestine and
are fundamental players in nutrient absorption with an additional role in host defense [1–4].
Key to the aforementioned processes is the brush border, so named because the apical surface
of the enterocytes is coated by thousands of microvilli of similar length and size organized in a
hexagonal arrangement [5]. The organization and structure of the microvilli is maintained by
a network of cytoplasmic and transmembrane proteins known as the intermicrovillar adhesion

6OAE, 7N86) and additional raw images for bead aggregation assays available at Dryad (https://doi.org/10.5061/dryad.w0vt4b8sh).

**Funding:** This work was supported in part by the Ohio State University and NIH (NIDDK R01 DK095811 to M.J.T.). The funders had no role in study design, data collection and analysis, decision to publish, or preparation of the manuscript.

**Competing interests:** The authors have declared that no competing interests exist.

**Abbreviations:** CDH1, cadherin-1; CDH2, cadherin-2; CDH23, cadherin-23; CDHR2, cadherin-related family member 2; CDHR5, cadherin-related family member 5; EC, extracellular cadherin; IMAC, intermicrovillar adhesion complex; MAD, membrane adjacent domain; MLD, mucin-like domain; PCDH15, protocadherin-15; PCDH24, protocadherin-24; RMSD, root mean square deviation; RU, resonance unit; SIAS, Sequence Identity and Similarity Server; SMD, steered molecular dynamics; SPR, surface plasmon resonance.

complex (IMAC), which includes proteins with extracellular adhesive domains and cytoplasmic parts tethered to the actin cytoskeleton that forms the interior of the microvilli [6–11]. Perturbations of brush border function are often associated with disease [12–15], and, not surprisingly, IMAC components are associated to foodborne diarrhea, chronic atrophic gastritis, and cholangiocarcinoma, while disruption of the IMAC complex results in intestinal dysfunction [9,16–20].

The extracellular parts of the IMAC stem from a pair of cellular adhesion proteins that belong to the cadherin superfamily, protocadherin-24 (PCDH24, also known as cadherin-related family member 2 [CDHR2]) and mucin-like protocadherin (CDHR5) [9,21]. These 2 nonclassical protocadherins form intermicrovillar links essential for brush border morphogenesis and function [9,10], stabilizing the microvillar hexagonal patterns as shown by freeze-etch electron microscopy and PCDH24 immunolabeling combined with transmission electron microscopy [9]. Mutations that impair the PCDH24 and CDHR5 interaction, studied in knockdowns of PCDH24 and CDHR5, cause a remarkable reduction in microvillar clustering ex vivo [9]. A PCDH24 knockout mouse model was found to be viable, but body weight of mutant mice was lower than wild type, and there were defects in the packing of the microvilli in the brush border [8]. Despite the important physiological role played by PCDH24 and CDHR5, little is known about the molecular details of how these proteins interact to form the intermicrovillar links.

PCDH24 and CDHR5 are members of a large superfamily of proteins with extracellular domains that often mediate adhesion and that have contiguous and similar, but not identical, extracellular cadherin (EC) repeats. The linker regions between the EC repeats typically bind 3 calcium ions essential for adhesive function [22–25]. PCDH24 belongs to the Cr-2 subfamily and has 9 EC repeats, a membrane adjacent domain (MAD10), a transmembrane domain, and a C-terminal cytoplasmic domain [21,26–30]. CDHR5 belongs to the Cr-3 subfamily and has 4 EC repeats, a mucin-like domain (MLD), a transmembrane domain, and a C-terminal cytoplasmic domain [31]. Some human CDHR5 isoforms lack the MLD, which is not essential for the interaction with human PCDH24 [9]. Sequence alignments of the N-terminal repeats (EC1-3) suggest that PCDH24 and CDHR5 are similar to another pair of heterophilic interacting cadherins: Cadherin-23 (CDH23) and protocadherin-15 (PCDH15), the cadherins responsible for the formation of inner-ear tip links [32–36]. PCDH24 is most similar to CDH23, having the residues and elongated N-terminal β-strand that are predicted to favor the formation of an atypical calcium-binding site at its tip [33,34]. CDHR5 is most similar to PCDH15, and features cysteine residues predicted to form a disulfide bond at its tip [32,36]. In addition, a mutation in CDHR5, p.R84G (residue numbering throughout the text corresponds to processed proteins, see Methods), which mimics a deafness-related mutation in PCDH15, interferes with the intermicrovillar links formed by PCDH24 and CDHR5 [9], suggesting that these proteins interact in a similar fashion to the heterophilic tip-link "handshake" formed by CDH23 and PCDH15 [32,36,37].

To better understand how PCDH24 and CDHR5 interact to form intermicrovillar links, we used a combination of structural and bead aggregation assays to characterize the adhesive properties and mechanisms of these cadherin family members. We present the X-ray crystallographic structures of *Homo sapiens* (*hs*) PCDH24 EC1-2 in 2 distinct forms, as well as of *Mus musculus* (*mm*) PCDH24 EC1-3 and *hs* CDHR5 EC1-2. The structures give insight into possible binding mechanisms, which are further probed by bead aggregation assays revealing that the human and mouse intermicrovillar cadherins do not engage in the same homophilic and heterophilic interactions. Based on these results, we suggest models for the PCDH24 homophilic complex and the PCDH24/CDHR5 heterophilic interaction utilized to form intermicrovillar links.

## Results

### PCDH24 and CDHR5 tip sequences are poorly conserved across species

Intermicrovillar links formed by PCDH24 and CDHR5 in the enterocyte brush border are similar to inner-ear tip links formed by CDH23 and PCDH15. Both types of links are essential for the development, assembly, and function of actin-based structures (microvilli in the gut and stereocilia in the inner ear), and both are made of long nonclassical cadherin proteins that have similar cytoplasmic partners involved in interactions with the cytoskeleton and in signaling [10,38]. Sequence analyses have revealed similarities between the tips of PCDH24 and CDH23 and between the tips of CDHR5 and PCDH15 [32]. Given that CDH23 and PCDH15 engage in a heterophilic "handshake" complex involving their EC1-2 tips, it has been proposed that PCDH24 and CDHR5 might use a similar binding mechanism [9,32]. To further explore this hypothesis, we performed sequence analyses comparing the tips of these cadherins to each other, to classical cadherins, and across species.

Interestingly, alignments of EC repeat sequences across 13 species for intermicrovillar and tip-link cadherins reveal poor conservation for PCDH24 and CDHR5 when compared to CDH23 and PCDH15. Average percent identity computed from these alignments is 50.7% for CDH23 EC repeats and 49.5% for PCDH15 repeats, with the N- and C-terminal ends being more conserved than the middle EC repeats [39,40]. In contrast, average percent identity is 16.6% for PCDH24 and 10.3% for CDHR5, considerably lower when compared to values for CDH23 and PCDH15. Moreover, the N- and C-terminal ends of PCDH24 and CDHR5 tend to be less conserved than the middle region of these proteins (Fig 1A, S1 and S2 Tables, and S1 and S2 Figs). Nevertheless, multiple sequence alignments of PCDH24 and CDH23 EC1 repeats confirm that PCDH24 has an elongated N-terminus that should facilitate the formation of calcium-binding site 0 as observed in structures of CDH23 (S3A Fig). Binding of calcium at this site in CDH23 is mediated by several acidic residues, which are also present and conserved in PCDH24. Similarly, sequence alignments of CDHR5 and PCDH15 EC1 repeats confirm that CDHR5 has the conserved cysteine residues that form a stabilizing disulfide bond at the tip of PCDH15 EC1 (S3B Fig). None of these proteins feature the tryptophan residues that mediate homophilic binding in classical cadherins [41–43], further supporting the hypothesis that PCDH24 and CDHR5 might form a tip-link-like handshake complex mediating adhesion.

Pairwise sequence identity computed across 5 species for the N-terminal repeats (EC1-3) of PCDH24, CDHR5, CDH23, PCDH15, and the classical cadherins E-cadherin (CDH1) and N-cadherin (CDH2), also show different trends of sequence conservation (Fig 1B, S3 Table). Average percent identity in pairwise comparisons of EC1-3 sequences is high for CDH2 (89%), CDH23 (86%), and PCDH15 (82%); moderate for CDH1 (61%); and lowest for PCDH24 (49%) and CDHR5 (38%). This is evident when comparing sequences from the most evolutionary distant species (human and fish). For instance, *hs* and *Danio rerio* (*dr*) PCDH24 EC1-3 sequences are 39% identical, while *hs* and *dr* CDH1 EC1-3 sequences are 54% identical. Sequence differences are still large for the human and mouse PCDH24 and CDHR5 protein tips, with 75% identity for the *hs* and *mm* PCDH24 EC1-3 sequences and 66% identity for the *hs* and *mm* CDHR5 EC1-3 sequences (compared to CDH1, CDH2, CDH23, and PCDH15 EC1-3 sequences with *hs* and *mm* pairwise identities between 83% and 98%) (Fig 1B). Proteins with sequence identity as low as 30% can share similar folds [44], yet the low PCDH24 and CDHR5 sequence conservation in multiple sequence alignments and pairwise comparisons suggests that details of their structures and function differ across species.

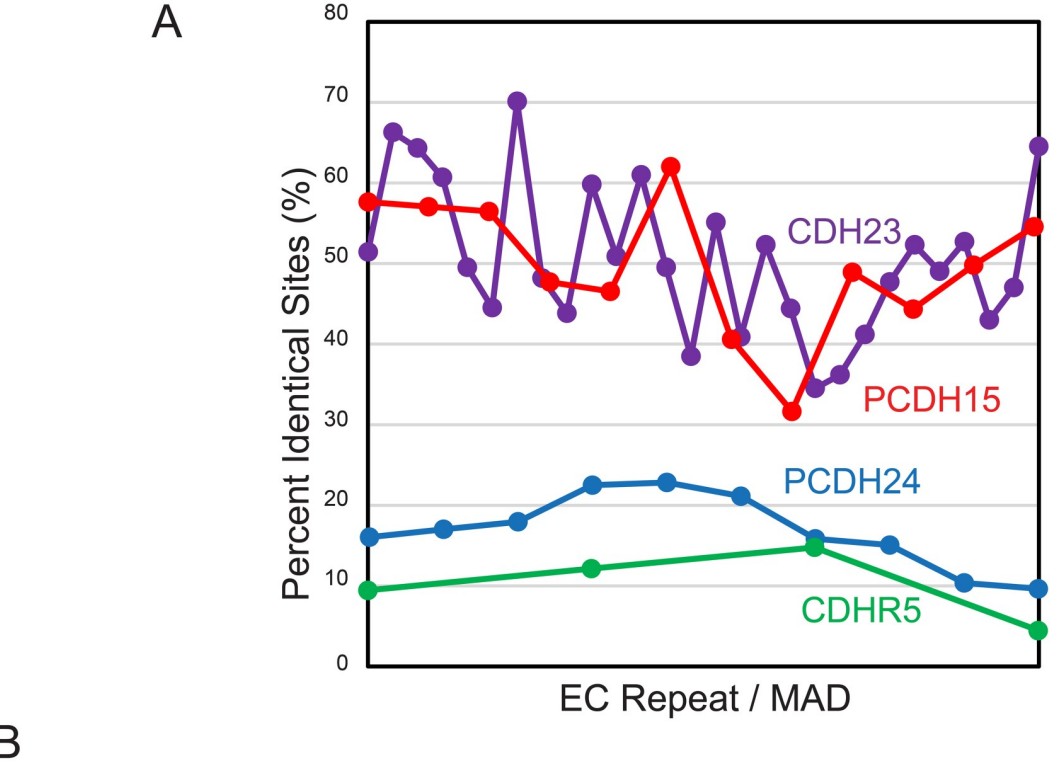

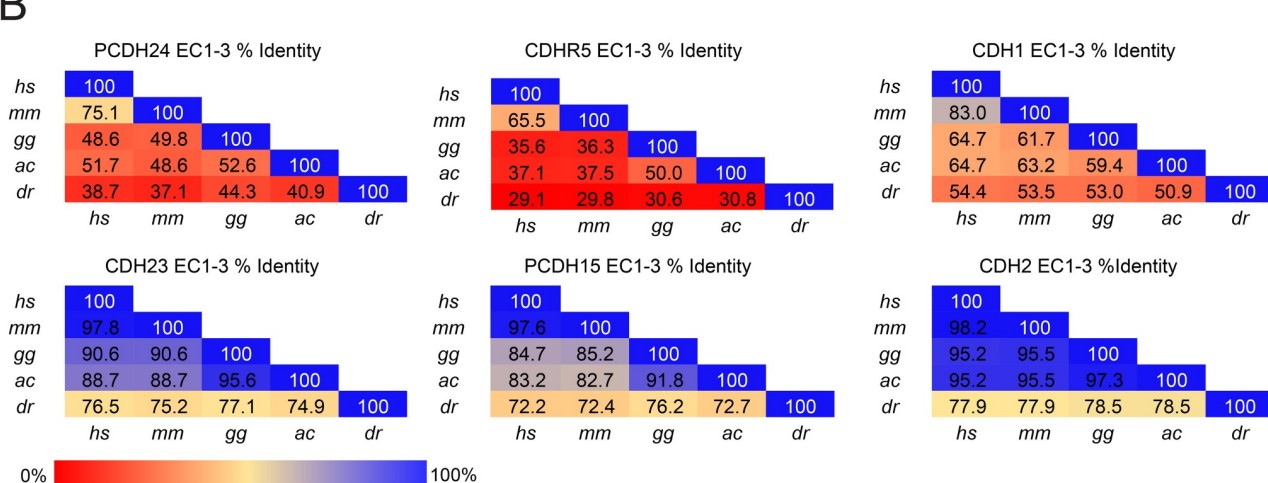

**Fig 1. Comparison of sequence conservation across species.** (**A**) Percent identity across the extracellular domains of PCDH24 (EC1-MAD10), CDH23 (EC1-MAD28), CDHR5 (EC1-4), and PCDH15 (EC1-MAD12) plotted against EC repeat/MAD number. Overall, the cadherins of the inner-ear tip link, CDH23 and PCDH15, have higher identity across a variety of species, while the intermicrovillar-link cadherins, PCDH24 and CDHR5, have poor sequence conservation across species (S1 and S2 Tables and S1 and S2 Figs). N- and C-termini are more conserved in CDH23 and PCDH15 than the middle region of these proteins. An opposite trend is observed for PCDH24 and CDHR5. (**B**) Percent identity of the first 3 EC repeats of PCDH24, CDHR5, CDH23, PCDH15, CDH1, and CDH2 demonstrate that PCDH24 and CDHR5 sequences are not highly conserved across several species (*Homo sapiens*, hs; *Mus musculus*, mm; *Gallus gallus*, gg; *Anolis carolinesis*, ac; *Danio rerio*, dr). However, the N-terminal repeats of CDH23, PCDH15, CDH1, and CDH2 are highly conserved. Sequences were obtained from NCBI (S3 Table and Methods). CDH1, Cadherin-1; CDH2, Cadherin-2; CDH23, Cadherin-23; CDHR2, cadherin-related family member 2; CDHR5, cadherin-related family member 5; EC, extracellular cadherin; MAD, membrane adjacent domain; PCDH15, protocadherin-15; PCDH24, protocadherin-24.

## Structures of human and mouse PCDH24 tips reveal N-terminal conformational variability

Previous bead aggregation assays using the *hs* PCDH24 protein revealed that its full-length extracellular domain can mediate adhesion through homophilic interactions with itself and through heterophilic interactions with *hs* CDHR5 [9]. Members of the cadherin superfamily of proteins often engage in homophilic and heterophilic interactions using their N-terminal tips (EC1 for classical cadherins; EC1-2 for tip-link cadherins; and EC1-4 for clustered, δ1, and δ2 protocadherins) [22,36,52,53,41,45–51]. Therefore, to gain insights into the molecular basis of PCDH24-mediated adhesion and to explore the consequences of poor sequence conservation across species, we solved the crystal structures of *hs* PCDH24 EC1-2 in 2 arrangements (forms I and II) refined at 2.3 Å and 3.2 Å resolution, as well as of *mm* PCDH24 EC1-3 refined at 2.1 Å resolution (Table 1). These structures, obtained from crystals of proteins refolded from

**Table 1. Statistics for PCDH24 and CDHR5 structures.**

| Data Collection and Refinement | *Hs* PCDH24 EC1-2 I | *Hs* PCDH24 EC1-2 II | *Mm* PCDH24 EC1-3 | *Hs* CDHR5 EC1-2 |
|---|---|---|---|---|
| Space Group | $P2_1$ | $P2_1$ | $P321$ | $I222$ |
| Unit cell parameters | | | | |
| • *a, b, c* (Å) | 270.9, 119.8, 74.3 | 63.9, 86.6, 104.9 | 74.8, 74.8, 140.0 | 50.4, 78.4, 121.4 |
| • α, β, γ (°) | 90, 104.12, 90 | 90, 103.46, 90 | 90, 90, 120 | 90, 90, 90 |
| Molecules per asymmetric unit | 4 | 4 | 1 | 1 |
| Beam Source | APS 24-ID-C | APS 24-ID-E | APS 24-ID-C | APS 24-ID-E |
| Wavelength (Å) | 0.97910 | 0.97918 | 0.97910 | 0.97918 |
| Resolution limit (Å) | 2.3 | 3.2 | 2.1 | 1.9 |
| Unique Reflections | 50,945 (2,128) | 18,809 (810) | 27,040 (1,237) | 19,398 (943) |
| Redundancy | 3.1 (2.6) | 3.5 (2.5) | 8.60 (5.2) | 24.10 (21.8) |
| Completeness (%) | 95.5 (80.3) | 96.3 (89.1) | 99.2 (90.6) | 99.7 (99.7) |
| Average I/σ (I) | 10.9 (3.0) | 5.6 (2.6) | 27.0 (4.4) | 46.14 (14.5) |
| $R_{merge}$ | 0.105 (0.255) | 0.257 (0.506) | 0.078 (0.337) | 0.074 (0.263) |
| **Refinement** | | | | |
| Resolution range (Å) | 50–2.3 (2.34–2.30) | 50–3.2 (3.26–3.18) | 50–2.1 (2.14–2.10) | 50–1.9 (1.93–1.90) |
| Residues (atoms) | 861 (6,610) | 848 (6,500) | 318 (2,447) | 204 (1,656) |
| Water Molecules | 238 | 33 | 98 | 128 |
| $R_{work}$ (%) | 20.5 (27.8) | 21.4 (23.5) | 21.0 (34.3) | 19.7 (20.9) |
| $R_{free}$ (%) | 23.5 (33.0) | 29.0 (31.3) | 24.3 (39.6) | 23.4 (25.9) |
| Rms deviations | | | | |
| • Bond lengths (Å) | 0.013 | 0.017 | 0.014 | 0.013 |
| • Bond angles (°) | 1.460 | 2.020 | 1.632 | 1.652 |
| B factor average | | | | |
| • Protein | 41.00 | 44.33 | 64.31 | 30.53 |
| • Ligand/ion | 36.90 | 50.44 | 51.49 | 39.42 |
| • Water | 31.65 | 11.78 | 50.99 | 33.60 |
| **Ramachandran plot regions** | | | | |
| Most favored (%) | 91.5 | 86.8 | 90.7 | 91.2 |
| Additionally allowed (%) | 8.3 | 12.8 | 8.9 | 8.8 |
| Generously allowed (%) | 0.3 | 0.4 | 0.4 | 0.0 |
| Disallowed (%) | 0.0 | 0.0 | 0.0 | 0.0 |
| **PDB ID code** | 5CZR | 7N86 | 5CYX | 6OAE |

CDHR5, cadherin-related family member 5; PCDH24, protocadherin-24.

*Escherichia coli* (*E. coli*) inclusion bodies or expressed in mammalian cells (see Methods), revealed both expected and unexpected architectural features.

The 2 *hs* PCDH24 EC1-2 structures have 4 molecules in the asymmetric unit including residues N1 to L217 (form I) and residues N1 to D218 (form II) for each monomer. The *mm* PCDH24 EC1-3 structure has 1 molecule in the asymmetric unit including residues N1 to D328. All structures had good-quality electron density maps that allowed the unambiguous positioning of side chains for most of the residues and of glycosylation sugars for *hs* PCDH24 EC1-2 II (S4A–S4C Fig). Comparisons among *hs* PCDH24 EC1-2 monomers within each of the 2 crystal structure forms reveal high structural similarity, with root mean square deviation (RMSD) values for main-chain atom comparisons ranging from 0.4 Å to 0.8 Å in form I and from 0.5 Å to 1.0 Å in form II. Similarly, RMSD values for comparisons among *hs* PCDH24 EC1-2 monomers from the 2 different structures range between 1.3 Å and 2.4 Å, indicating that overall fold is not dependent on glycosylation or method of protein production. As expected, all EC repeats in all 3 PCDH24 structures adopt the typical 7 β-strand Greek-key fold seen in extracellular domains of cadherin proteins, with β-strands labeled A to G for each repeat (Fig 2A–2D). Both *hs* and *mm* PCDH24 EC2 repeats have extended F and G strands, which are linked by a disulfide bond (C187:C201) that stabilizes an unusually long FG β-hairpin near the EC1-2 linker (Fig 2A–2D, S4A Fig).

The PCDH24 EC1-2 linker region is canonical in both human and mouse structures and features either calcium or sodium ions bound to sites 1 of distinct monomers, and calcium ions bound to sites 2 and 3 in all EC1-2 linker regions. These ions are coordinated by acidic residues from the cadherin motif NTerm-XEX-DXD-D(R/Y)(D/E)-XDX-DXNDN-CTerm (Fig 2A–2E, S5 Fig). Interestingly, the structure of mouse PCDH24 has a noncanonical calcium-binding linker between repeats EC2 and EC3, with a straight conformation and only 2 bound calcium ions (Fig 2C, 2D and 2F, S5 Fig). This linker lacks necessary conserved residues to bind the calcium or sodium ion that would occupy site 1 as the canonical DXE sequence is replaced with residues 172SYN174. Although the canonical DXNDN sequence motif is 213DXPDL217 at the EC2-3 linker, typical coordination of calcium at site 3 by the asparagine carboxamide oxygen is replaced by coordination through the backbone carbonyl group of a glutamate residue in our structure (p.E214 in mouse, p.Q214 in human). The last residue of the motif (typically asparagine, replaced here by p.L217) coordinates calcium at site 3 through its backbone carbonyl group, an interaction that is unchanged by residue variability at this position. Sequence conservation across species for the DXE/SYN and DXNDN/DXPDL motifs suggest that most species have a noncanonical linker region between repeats EC2 and EC3 with 2 bound calcium ions as observed in our *mm* PCDH24 EC1-3 structure. While noncanonical linker regions with 2 or less bound calcium ions have been shown to be flexible [54–56], equilibrium and steered molecular dynamics (SMD) simulations of *mm* PCDH24 EC1-3 show that rigidity and mechanical strength are not compromised in this noncanonical linker (S6 Fig, S8 Data), which might be relevant for potential interactions with binding partners and for its structural function in microvilli morphogenesis and maintenance.

As expected for members of the Cr-2 subfamily of nonclustered protocadherins, the *hs* PCDH24 EC1-2 structure shows an occupied site 0 calcium-binding site at the tip of EC1, with calcium-coordinating residues similar to those observed in structures of CDH23 (1N, 32DXDXD36, 80XDX$^{TOP}$; Fig 2A, 2B and 2G, S7A Fig). Unexpectedly, the *mm* PCDH24 EC1-3 structure shows an unoccupied site 0, with an extended N-terminus protruding away from EC1 as observed for classical cadherins that exchange their N-terminal strands to form adhesive bonds (Fig 2C, 2D and 2H, S7B Fig). The sequence motifs involved in calcium coordination at site 0 are conserved, suggesting that crystallization conditions, including 3M LiCl, may have facilitated the opening of this site. Alternatively, other differences between mouse and

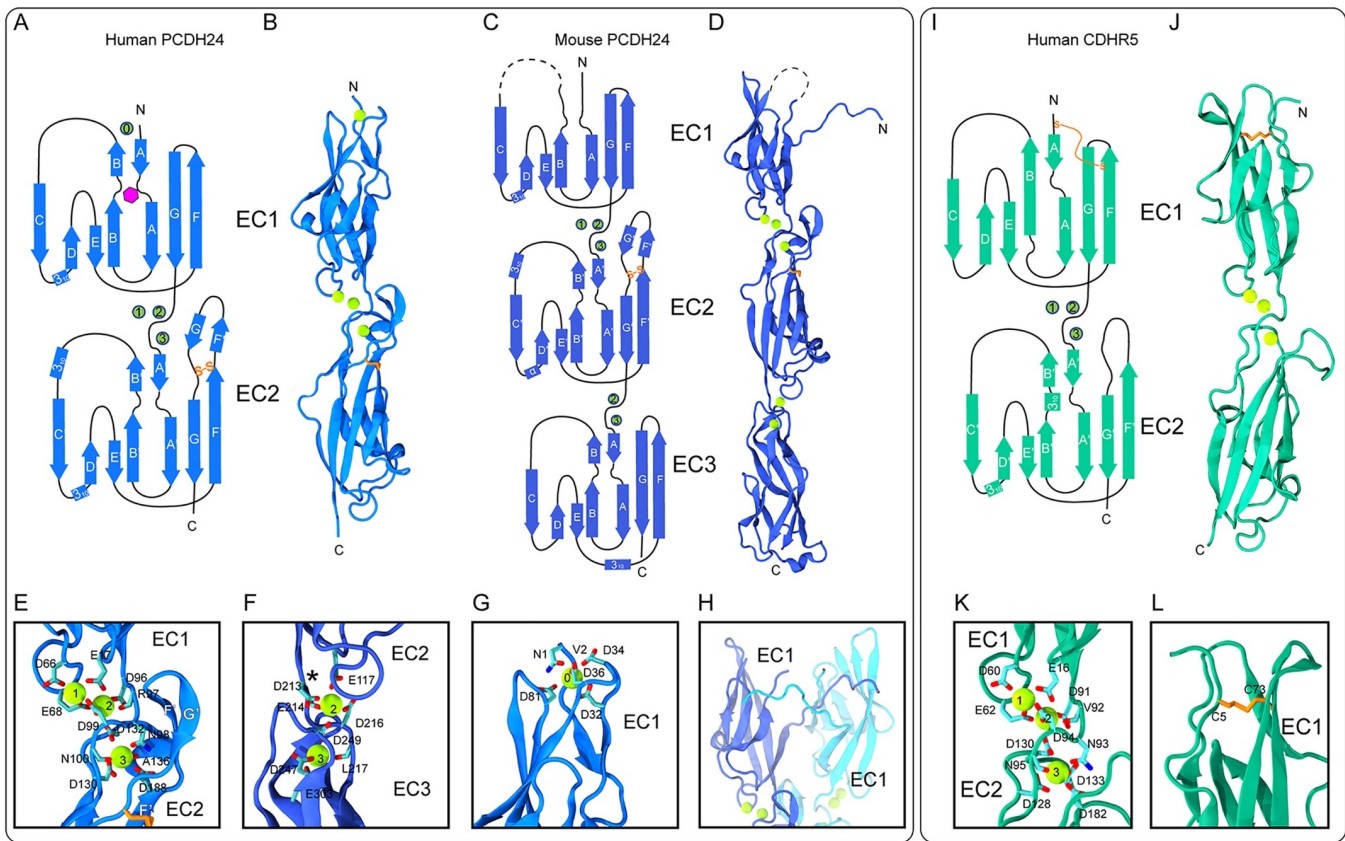

**Fig 2. Structures of human (*hs*) PCDH24 EC1-2, mouse (*mm*) PCDH24 EC1-3, and *hs* CDHR5 EC1-2.** (**A**) Topology of *hs* PCDH24 EC1-2. The EC1 and EC2 repeats have a typical cadherin fold with 7 β-strands. A disulfide bond in EC2 is highlighted in orange. Calcium ions are shown as green circles. N-linked glycosylation at site p.N9 is highlighted by a magenta hexagon. (**B**) A ribbon representation of *hs* PCDH24 EC1-2 with calcium ions in green. (**C**) Topology of the *mm* PCDH24 EC1-3 shown as in (A). The dashed line indicates a loop not resolved. (**D**) A ribbon representation of *mm* PCDH24 EC1-3 as in (B). The structure is similar to *hs* PCDH24 EC1-2, but the N-terminus projects away from EC1. A N-terminal calcium ion was not present. (**E**) Detail of a linker between EC1 and EC2 of *hs* PCDH24 I showing canonical calcium-binding sites. Some sidechain and backbone atoms are not shown for clarity. (**F**) Detail of the *mm* PCDH24 EC2-3 linker, which lacks several canonical calcium-binding residues and thus only has 2 calcium ions bound. Shown as in (E). An asterisk (*) indicates the position where a calcium residue would be found in the canonical linker. (**G**) Detail of the N-terminal calcium bound at the tip of EC1 in the *hs* PCDH24 EC1-2 I structure. A similar calcium-binding motif is seen at the tip of CDH23. Residues coordinating calcium are shown in stick with backbone atoms omitted. (**H**) Detail of the N-terminus of *mm* PCDH24 EC1-3 (blue) showing its interaction with another protomer (cyan) in the asymmetric unit. (**I**) Topology of *hs* CDHR5 EC1-2 shown as in (A). (**J**) A ribbon representation of *hs* CDHR5 EC1-2 with calcium ions in green. (**K**) Detail of the linker between EC1 and EC2 of *hs* CDHR5 showing canonical calcium-binding sites. Shown as in (I, J). (**L**) Detail of the disulfide bond in EC1 in the *hs* CDHR5 EC1-2 structure. A similar disulfide bond is seen in PCDH15 EC1. CDH23, Cadherin-23; CDHR5, cadherin-related family member 5; PCDH15, protocadherin-15; PCDH24, protocadherin-24.

human sequences might be responsible for the structural divergence. Most notably, residue R86 near the FG loop and the XDX$^{TOP}$ motif of *mm* PCDH24 EC1 seems to be incompatible with a closed N-terminal conformation and an occupied calcium-binding site 0. The open and closed conformations observed for PCDH24 EC1 will likely determine the type of homophilic or heterophilic interactions that mediate adhesion.

## Crystal contacts in human and mouse PCDH24 structures suggest distinct adhesive interfaces

Crystallographic contacts observed in cadherin structures have revealed multiple physiologically relevant interfaces [36,40,45,46,48,49,57]. The crystallographic packings observed for the *hs* and *mm* PCDH24 structures show various interfaces that further highlight differences

across species. The asymmetric unit of the *hs* PCDH24 EC1-2 I structure shows 4 molecules with 2 similar antiparallel *trans* dimers (S8A Fig) arranged perpendicular to one another to form a dimer of dimers (S8B Fig). Each antiparallel dimer positions EC1 from 1 monomer in front of EC2 from the next monomer, slightly shifted with respect to each other and with the extended FG β-hairpins near the EC1-2 linker regions mediating the interaction head-to-head. The antiparallel dimers have interface areas of 979 Å$^2$ and 939 Å$^2$ for monomers A:D and B:C, respectively. These values are both larger than an empirical threshold (856 Å$^2$) that distinguishes biologically relevant interactions from crystallographic packing artifacts [58]. Two other interfaces are small and unlikely to be biologically relevant (S8C and S8D Fig). Previous data have shown that *hs* PCDH24 mediates homophilic *trans* adhesion when *hs* CDHR5 is not present [9]. It is possible that the antiparallel EC1-2 interface seen in the *hs* PCDH24 EC1-2 I structure facilitates homophilic adhesion mediated by *hs* PCDH24, yet our SMD simulations suggest that this interface is weak (S9A Fig, S9 Data).

Interestingly, the *hs* PCDH24 EC1-2 II structure also has 4 molecules in the asymmetric unit, but these arrange differently and form a distinct set of interfaces. Remarkably, there are 2 sets of potential antiparallel *trans* dimers (S10 Fig). In the first one, the EC1 repeat from 1 monomer is in front of EC2 from the next monomer, slightly shifted and with glycosylation sugars at p.N9 residues and the extended FG β-hairpins near the EC1-2 linker regions all pointing away from the interface (S10A Fig). These antiparallel dimers have interface areas of 1,023 Å$^2$ and 1,019 Å$^2$ for monomers A:B and C:D, respectively, and feature an overlap of aromatic rings contributed by p.Y67 and p.Y71 residues that might be critical for binding. In the second set of antiparallel *trans* dimers, the EC1 repeat from 1 monomer is also positioned in front of EC2 from the next monomer, but at an angle that reduces their contacts, gives space for sugars stemming from p.N9, and that favors overlap of the extended FG β-hairpins near the EC1-2 linker regions (S10B Fig). These antiparallel dimers have interface areas of 418 Å$^2$ and 436 Å$^2$ for monomers A:C and B:D, respectively. The 2 sets of antiparallel *trans* dimers in the *hs* PCDH24 EC1-2 II structure come together to form an asymmetric unit in which various other smaller interfaces are observed (S10C–S10G Fig). It is possible that the largest antiparallel EC1-2 interface seen in the *hs* PCDH24 EC1-2 II structure (A:B and C:D) facilitates homophilic adhesion mediated by *hs* PCDH24. Predictions from SMD simulations show that this interface is stronger than the antiparallel interface observed in *hs* PCDH24 EC1-2 I when probed under the same conditions (S9C Fig, S10 Data), thus supporting the possible physiological relevance of the *hs* PCDH24 EC1-2 II arrangement.

In contrast to the crystal contacts observed in the *hs* PCDH24 EC1-2 I and II structures, the crystal packing of *mm* PCDH24 EC1-3 (space group *P*321) results in multiple large interfaces generated from rotations of a single molecule in the asymmetric unit. A parallel trimer is present around the axis of symmetry, with the EC1 N-termini from each protomer protruding away and inserting into pockets of adjacent EC1 repeats in a clockwise fashion, thereby interlocking the EC1 domains (S11A and S11B Fig). This is possible because of the open conformation observed for the mouse EC1 N-terminus without a bound calcium ion at site 0, and it is reminiscent of how the EC1 N-terminus of classical cadherins is exchanged resulting in the insertion of W2 into an hydrophobic binding pocket of the partner molecule to form a *trans* dimer (S7C and S7D Fig) [41,42]. Interestingly, a glycosylation site is predicted at p.N9, and sugars protruding from this site, as observed in the *hs* PCDH24 EC1-2 II structure, may interfere with or regulate the formation of interlocking EC1 repeats, as has been observed for other cadherins [57].

The interlocking of mouse EC1 repeats in the trimer facilitates the formation of 2 parallel *cis* interfaces between EC1-3 protomers involving the FG β-hairpin interacting with the neighboring EC2-3 linker. These *cis* interfaces are large with an interface area of 1,352.6 Å$^2$ between 2 monomers (S11A Fig). In addition, a fully overlapped antiparallel *trans* interface is observed

between EC1-3 protomers with an area of 1,221.2 Å$^2$ (S11C Fig). Sugars at a second predicted glycosylation site located in the center of the *cis* trimer (p.N161) and at a third predicted glycosylation site near the *trans* interface between EC1 and EC3 (p.N280) may interfere with or regulate the formation of these interfaces. Together, the *cis* trimer and *trans* dimer could form a glycosylation-modulated interlocking arrangement of PCDH24 molecules that is different from the *trans* interactions seen in the *hs* PCDH24 EC1-2 I and II structures. Sequence conservation mapped to the *mm* PCDH24 EC1-3 structure reveals that only core residues are highly conserved, highlighting poor conservation for surface exposed residues potentially involved in adhesive interactions (S12 Fig). Motivated by the sequence and structural differences observed for the human and mouse PCDH24 tips, we used bead aggregation assays to determine the minimum adhesive unit of PCDH24 homophilic binding in both species.

## Human, but not mouse, PCDH24 mediates homophilic adhesion

To determine which EC repeats are essential for *hs* and *mm* PCDH24 homophilic adhesion, we created a library of constructs encoding for their full-length extracellular domains (EC1-MAD10) and various truncations (EC1-7, EC1-6, EC1-5, EC1-4, EC1-3, EC1-2, and EC1), all fused to a C-terminal Fc tag. These protein fragments were produced in HEK293T cells and used for aggregation assays with Protein G magnetic beads (see Methods).

As expected, bead aggregation experiments using *hs* PCDH24 EC1-MAD10Fc showed large calcium-dependent aggregates (Fig 3A, 3L and 3M, S13A and S13I Fig, S1 Data). Similar large calcium-dependent aggregates were seen for *hs* PCDH24 EC1-7Fc through EC1-3Fc (Fig 3B–3F, 3L and 3M, S13B–S13F Fig, S1 Data). Further truncations indicate that smaller *hs* PCDH24 fragments, including just EC1, are capable of facilitating the formation of small bead aggregates that increase in size with rocking but never reach the larger sizes observed when using fragments with 3 or more EC repeats (Fig 3G, 3H and 3K–3M, S13G and S13H Fig). Controls performed using EDTA showed that our protein fragments are unable to mediate bead aggregation when calcium is depleted, except when using EC1Fc, possibly stabilized by the adjacent Fc tag (Fig 3I–3M, S13I Fig, S1 Data). These results suggest that *hs* PCDH24 mediates calcium-dependent homophilic adhesion with at least 3 distinct modes of adhesion mediated by *hs* PCDH24 EC1-3, *hs* PCDH24 EC1-2, and by *hs* PCDH24 EC1 alone. It is possible that each of these modes uses a different interface and that the antiparallel interfaces observed in our *hs* PCDH24 EC1-2 structures are low-affinity intermediate states that facilitate further interdigitation for the adhesive mode that drives the formation of larger bead aggregates.

To probe the most promising *trans* interface observed in our human PCDH24 structures and to understand the different modes of adhesion observed in our bead aggregation assays, we introduced mutations into *hs* PCDH24 EC1-2Fc and EC1-3Fc fragments and tested their adhesive properties. Residues p.Y67 and p.Y71, near the center of the largest antiparallel interface in the *hs* PCDH24 EC1-2 II structure (Fig 4A, S10 Fig), were mutated to alanine. Unexpectedly, bead aggregation experiments using *hs* PCDH24 EC1-2Fc p.Y67A and *hs* PCDH24 EC1-3Fc p.Y67A showed larger calcium-dependent aggregates than when using the wild-type protein fragments (Fig 4B, 4C, 4D, 4F and 4G, S2 Data). In contrast, bead aggregation experiments using *hs* PCDH24 EC1-2Fc p.Y71A did not show aggregation with results similar to EDTA controls (Fig 4H and 4I), while the size of the aggregates mediated by *hs* PCDH24 EC1-3Fc p.Y71A were markedly reduced when compared to those mediated by the wild-type protein (Fig 4C and 4E). Likely, the p.Y67A mutation introduces some flexibility into the linker region that strengthens the π–π interaction between p.Y71 residues from adjacent monomers, while the p.Y71A mutation impairs the formation of an adhesive interface that is essential for

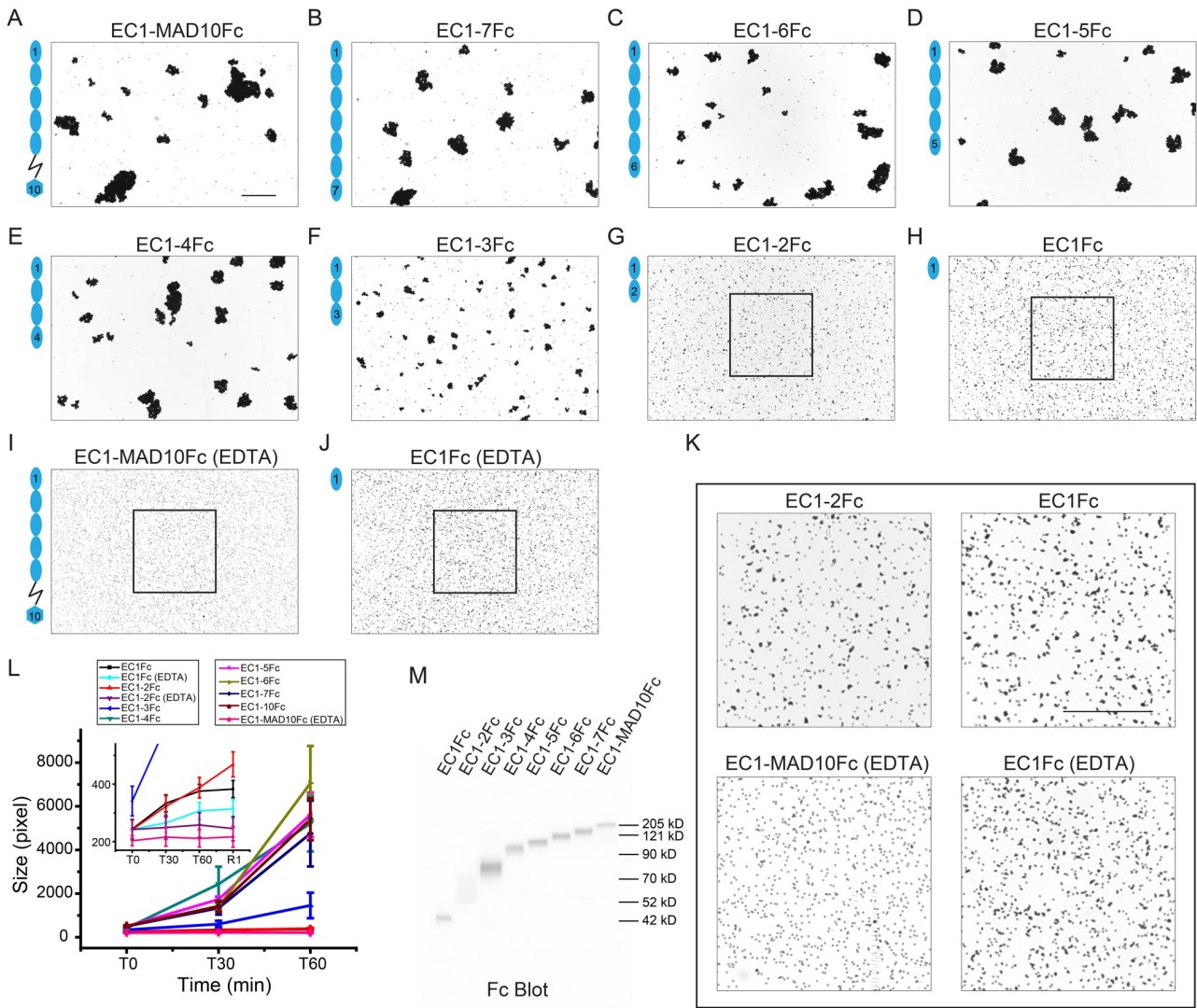

**Fig 3. Homophilic binding assays of *hs* PCDH24.** (**A-H**) Protein G beads coated with the Fc-tagged full-length *hs* PCDH24 extracellular domain (A) and its C-terminal truncation versions (B-H). Each image shows bead aggregation observed after 60 min in the presence of 2 mM CaCl$_2$. (**I, J**) Protein G beads coated with the Fc-tagged full-length *hs* PCDH24 extracellular domain (I) and EC1 (J) in the presence of 2 mM EDTA, shown as in (A-H). (**K**) Detail of binding assay results for EC1-2Fc, EC1Fc, EC1-MAD10Fc with EDTA, and EC1Fc with EDTA. Aggregation is still present for the shortest constructs. Bar– 500 μm. (**L**) Aggregate size for full-length and truncated versions of *hs* PCDH24 extracellular domains at the start of the experiment (T0) and after 30 and 60 min (T30, T60). Inset shows aggregate size of the shortest constructs compared to the EDTA control of EC1-MAD10Fc. An additional 1-min rocking step is denoted by R1. Error bars are standard error of the mean (*n* indicated in S4 Table; S1 Data). (**M**) Western blot shows expression and secretion of full-length and truncated versions of the Fc-tagged *hs* PCDH24 extracellular domain (S1 Raw Images). PCDH24, protocadherin-24.

bead aggregation mediated by *hs* PCDH24 EC1-2Fc and important for the larger bead aggregates mediated by *hs* PCDH24 EC1-3Fc. These results suggest that the largest antiparallel interface observed in the *hs* PCDH24 EC1-2 II structure is biologically relevant.

Given the low sequence identity between the human and mouse N-terminal EC repeats of PCDH24 when compared to CDH23, and the structural differences observed between *hs* PCDH24 EC1-2 and *mm* PCDH24 EC1-3, bead aggregation assays were also performed with the full-length extracellular domain of *mm* PCDH24. Bead aggregation was not observed for *mm* PCDH24 EC1-MAD10Fc (Fig 5A, 5B, 5E and 5G, S3 Data). These results suggest that the

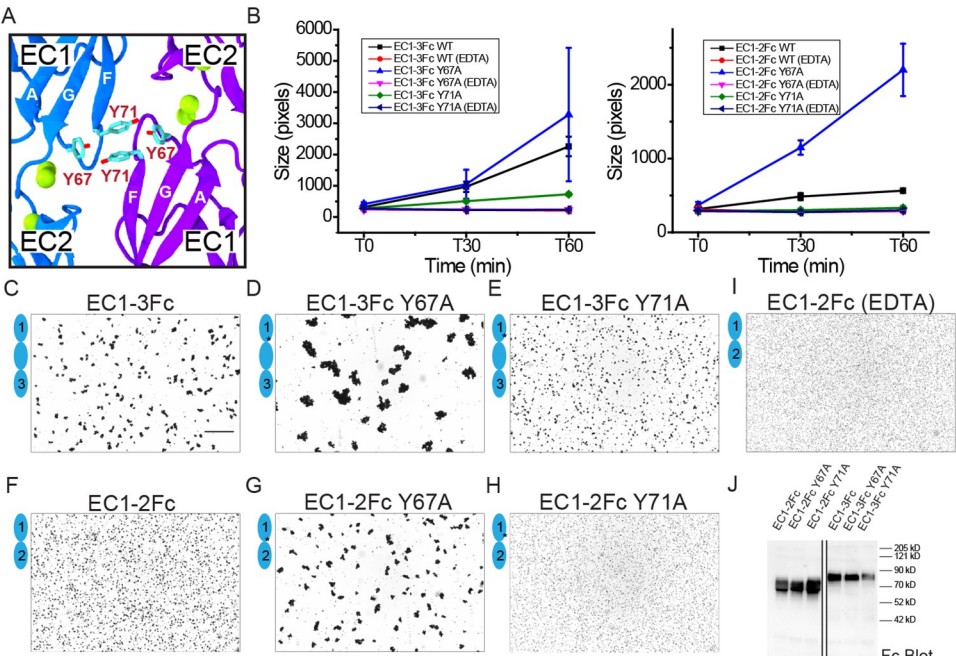

**Fig 4. Homophilic binding assays of *hs* PCDH24 mutants.** (**A**) Location of Y67 and Y71 residues at the largest antiparallel interface observed in the crystal structure *hs* PCDH24 EC1-2 II. Calcium ions are shown as green spheres, and EC1 strands are labeled. (**B**) Aggregate size for Fc-tagged *hs* PCDH24 EC1-3 (left) and EC1-2 (right) WT and mutants at the start of the experiment (T0) and after 30 and 60 min (T30, T60). Error bars are standard error of the mean (*n* indicated in S4 Table; S2 Data). (**C-H**) Protein G beads coated with *hs* PCDH24 EC1-3Fc WT (C), EC1-3Fc Y67A (D), EC1-3Fc Y71A (E), EC1-2Fc WT (F), EC1-2Fc Y67A (G), and EC1-2Fc Y71A (H). Each image shows bead aggregation observed after 60 min in the presence of 2 mM CaCl₂. (**I**) Protein G beads coated with *hs* PCDH24 EC1-2Fc WT in the presence of 2 mM EDTA, shown as in (C-H). Bar– 500 μm. (**J**) Western blot shows expression and secretion of Fc-tagged *hs* PCDH24 EC1-3 and EC1-2 WT and mutants (S1 Raw Images). PCDH24, protocadherin-24; WT, wild type.

large crystallographic antiparallel *trans* interface formed by *mm* PCDH24 EC1-3 (S11C Fig) does not mediate bead aggregation under the conditions tested and reveal that PCDH24 mediated homophilic adhesion is species dependent.

## Mouse, but not human, CDHR5 mediates homophilic adhesion

The full-length extracellular domain of *hs* CDHR5 was previously shown to not mediate homophilic adhesion [9]. The low sequence identity between the human and the mouse CDHR5, especially at their N-termini, and the species-dependent homophilic adhesive behavior of PCDH24, prompted us to test if *mm* CDHR5 could mediate adhesion. Full-length cadherin extracellular domains (without MLDs) of both human and mouse CDHR5 were used for bead aggregation assays. As expected, *hs* CDHR5 EC1-4Fc did not mediate bead aggregation in the presence and the absence of calcium. However, *mm* CDHR5 EC1-4Fc did mediate bead aggregation in the presence of calcium (Fig 5C, 5D, 5F and 5G, S4 Data). These results suggest that, like PCDH24, CDHR5 homophilic adhesiveness is species dependent.

To determine the minimal adhesive unit of *mm* CDHR5, a library of truncated cadherin fragments was generated and used for bead aggregation assays. Calcium-dependent bead aggregation was observed when using beads coated with *mm* CDHR5 EC1-4Fc and EC1-3Fc, but not when using EC1-2Fc and EC1Fc (Fig 6A–6D and 6G–6I, S14A–S14E Fig, S5 Data). Hence, *mm* CDHR5 EC1-3 is sufficient to mediate homophilic adhesion.

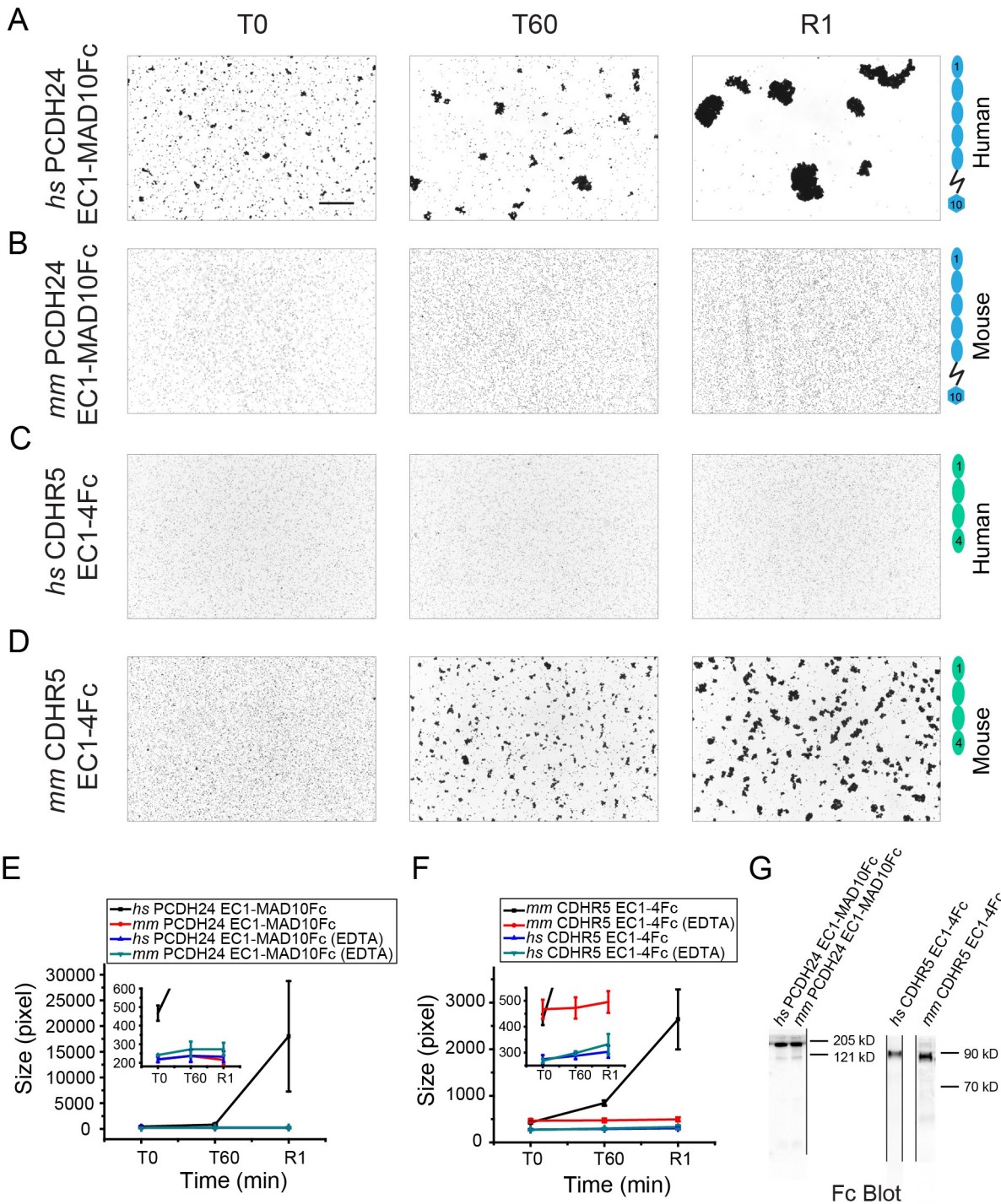

**Fig 5. Comparison of homophilic binding assays of *hs* and *mm* PCDH24 and CDHR5. (A-D)** Protein G beads coated with Fc-tagged full-length *hs* PCDH24 (A), full-length *mm* PCDH24 (B), full-length *hs* CDHR5 (C), and full-length *mm* CDHR5 (D) cadherin extracellular domains, including MAD10 for PCDH24. None of the CDHR5 fragments included the MLD. Images show the aggregation observed at the start of the experiment (T0), after 60 min (T60) followed by rocking for 1 min (R1) in the presence of 2 mM CaCl$_2$. Bar– 500 μm. (**E**) Aggregate size (S3 Data) for Fc-tagged full-length *hs* and *mm* PCDH24 extracellular domains in the presence of 2 mM CaCl$_2$ and 2 mM EDTA at the start of the experiment (T0), after 60 min (T60) followed by rocking for 1 min (R1). Inset shows aggregate size of Fc-tagged full-length *mm* PCDH24 extracellular domain compared to the EDTA control of Fc-tagged full-length *hs* and *mm* PCDH24 extracellular domains. (**F**) Aggregate size (S4 Data) for Fc-tagged full-length *hs* and *mm* CDHR5 cadherin extracellular domains in the presence of 2 mM CaCl$_2$ and 2 mM EDTA shown as in (E). Inset shows aggregate size of the Fc-tagged full-length *hs* CDHR5 cadherin extracellular domain compared to the EDTA control of Fc-tagged full-length *hs* and *mm* CDHR5 cadherin extracellular domain. Error bars in (E) and (F) are

standard error of the mean (*n* indicated in S4 Table). (**G**) Western blot shows expression and secretion of Fc-tagged full-length *hs* and *mm* PCDH24 and CDHR5 cadherin extracellular domains (S1 Raw Images). CDHR5, cadherin-related family member 5; MLD, mucin-like domain; PCDH24, protocadherin-24.

To provide insight into the interface used by *mm* CDHR5 to mediate homophilic adhesion, we tested mutations that alter a site similar to that mutated in *hs* CDHR5 (p.R84G) preventing heterophilic binding to *hs* PCDH24. Bead aggregation assays were performed with both *mm* CDHR5 EC1-4Fc p.R82G and p.E84G, and both mutations abolished bead aggregation mediated by *mm* CDHR5 EC1-4Fc in the presence of calcium (Fig 6E, 6F, 6H and 6I, S5 Data). These results suggest that the interface used for mediating heterophilic adhesion between PCDH24 and CDHR5 and the interface used for mediating homophilic adhesion by *mm* CDHR5 might be similar.

## Structure of human CDHR5 EC1-2 reveals a PCDH15-like fold and suggests adhesive interfaces

To gain insight into CDHR5's adhesive function, we solved the crystal structure of *hs* CDHR5 EC1-2 refined at 1.9 Å resolution (Table 1). This structure, obtained using protein expressed

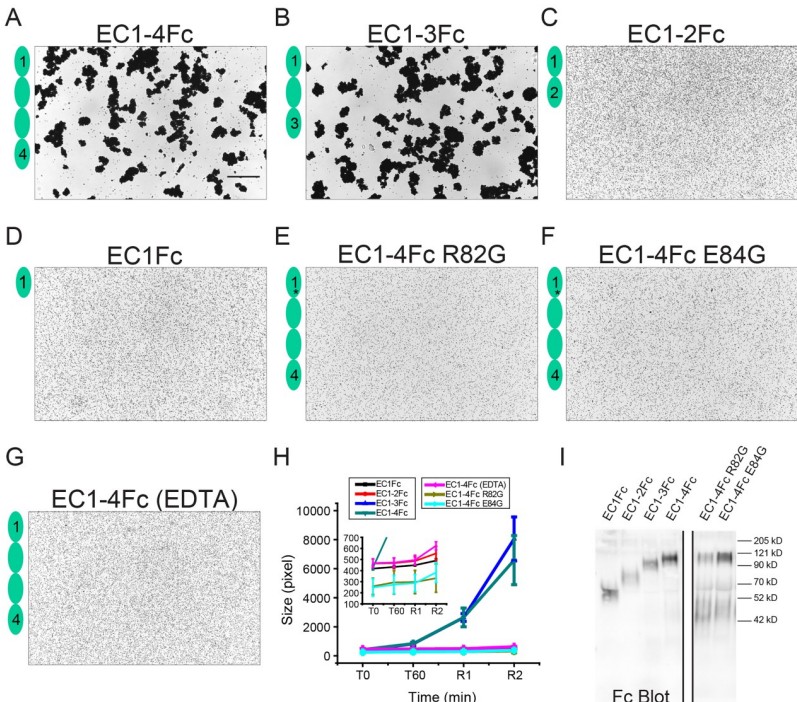

**Fig 6. Homophilic binding assays of *mm* CDHR5.** (**A-D**) Protein G beads coated with the Fc-tagged full-length *mm* CDHR5 cadherin extracellular domain (A) and its C-terminal truncation versions (B-D). Each image shows bead aggregation observed after 60 min followed by rocking for 2 min in the presence of 2 mM CaCl$_2$. (**E, F**) Protein G beads coated with mutants *mm* CDHR5 EC1-4Fc R82G (E) and EC1-4Fc E84G (F) observed after 60 min followed by rocking for 2 min in the presence of 2 mM CaCl$_2$. (**G**) Protein G beads coated with the full-length *mm* CDHR5 cadherin extracellular domain in the presence of 2 mM EDTA shown as in (A). Bar– 500 μm. (**H**) Aggregate size for full-length, truncated versions and mutants of *mm* CDHR5 cadherin extracellular domains at the start of the experiment (T0), after 60 min (T60) followed by rocking for 1 min (R1) and 2 min (R2). Inset shows aggregate size of EC1Fc, EC1-2Fc, the EDTA control of EC1-4Fc and EC1-4Fc mutants R82G and E84G. Error bars are standard error of the mean (*n* indicated in S4 Table; S5 Data). (**I**) Western blot shows expression and secretion of Fc-tagged full-length, truncated, and mutant versions of *mm* CDHR5 cadherin extracellular domains (S1 Raw Images). CDHR5, cadherin-related family member 5.

in *E. coli* and refolded from inclusion bodies (see Methods), had 1 molecule in the asymmetric unit including residues Q1 to L207, with clear electron density for residues A2 to A205 (Fig 2I and 2J, S4D Fig). Like other cadherins, the structure of *hs* CDHR5 EC1-2 shows straight consecutive EC repeats, each with a typical 7 β-strand Greek-key fold (Fig 2I and 2J). The linker region between the repeats has the canonical acidic residues that coordinate calcium and 3 calcium ions bound to it (Fig 2K, S15 Fig). Similar to PCDH15, a member of the Cr-3 subfamily of nonclustered protocadherins, *hs* CDHR5 EC1 has a disulfide bridge between cysteine residues C5 and C73 on strands A and F, respectively (Fig 2L, S4D and S7E Figs). This disulfide bond keeps the N-terminal strand tucked to the protomer in a closed conformation, reminiscent of how calcium at site 0 facilitates closing of the N-terminal strand in EC1 repeats of CDH23 and human PCDH24.

Next, we analyzed crystallographic contacts in the *hs* CDHR5 EC1-2 structure (S16 Fig), which could lead to an understanding of how mouse, but not human, CDHR5 mediates homophilic adhesion and how the heterophilic complex with PCDH24 is formed. The 2 largest contacts within the *hs* CDHR5 EC1-2 crystal structure are interfaces in antiparallel arrangements. The largest, with an interface area of 1,433.9 Å$^2$, is formed by interactions between the EC1 repeat of 1 protomer and EC2 of the adjacent one, with a slight shift in the register of the antiparallel dimeric interface (S16A Fig). There are up to 3 predicted N-glycosylation sites at this interface in the sequences for the human (p.N19) and the mouse (p.N17 and p.N107) proteins. The next largest contact involves interactions between antiparallel EC2 repeats facing each other (S16B Fig), with an interface area of 530.2 Å$^2$, small relative to likely physiologically relevant interfaces (>856 Å$^2$), but comparable to interface area per EC repeat reported for physiologically relevant contacts in protocadherins [45,46,59]. In this arrangement, the EC3 repeat (not present in the structure) would interact with EC1, thus increasing the total contact area. None of the predicted N-glycosylation sites for the human or mouse sequences are at the EC2 interface of this *trans* dimer. An X-shaped crystallographic contact is also present (S16C Fig), with a small interface area (364.7 Å$^2$) involving EC1 to EC1 interactions. Additional smaller interfaces are present but are unlikely to be physiologically relevant given their atypical arrangement and small interface areas of 347.9 Å$^2$, 132.7 Å$^2$, and 114.0 Å$^2$, respectively (S16D–S16F Fig).

Although *hs* CDHR5 does not mediate adhesion in bead aggregation assays, some of the contacts discussed above can serve as templates to predict the interfaces that could be used by *mm* CDHR5 to mediate homophilic adhesion. The largest, antiparallel EC1-2 interface might be impaired by glycosylation at a site predicted to exist in the mouse sequence (p.N107) but not in the human protein (p.R107). In addition, bead aggregation assays suggest that repeat EC3 is needed for adhesive interactions, suggesting that this EC1-2 interface is not driving CDHR5 homophilic adhesion. In contrast, the antiparallel EC1-3 interface mediated by EC2 interactions is compatible with results from bead aggregation assays indicating that the first 3 N-terminal repeats are sufficient for *trans* adhesiveness, but details of the arrangement are likely to be different due to the absence of EC3 in our structure and the low sequence conservation of surface residues (S17 Fig). These contacts may also guide the search for interaction modes that facilitate the formation of PCDH24 and CDHR5 heterophilic complexes.

## Human and mouse heterophilic PCDH24 and CDHR5 bonds are distinct

Previous bead aggregation assays showed that full-length extracellular domains of human PCDH24 and CDHR5 mediate heterophilic adhesion [9]. We used our library of Fc-tagged PCDH24 and CDHR5 cadherin fragments to test for heterophilic adhesiveness for both human and mouse proteins. To distinguish PCDH24 from CDHR5 coated beads, we used

green and red fluorescent beads, respectively. We confirmed that human PCDH24 and CDHR5 cadherin extracellular domains mediate calcium-dependent heterophilic adhesion and found that the mouse proteins also facilitate calcium-dependent bead aggregation (Fig 7A, 7B and 7I–7K, S18A and S18B Fig, S6 and S7 Data), suggesting that the heterophilic intermicrovillar cadherin bond driving intestinal brush border assembly is species independent.

Next, we used truncated protein fragments and found that repeats EC1-4Fc of PCDH24 and CDHR5 from both species are sufficient to drive heterophilic bead aggregation (Fig 7A, 7B and 7I–7K, S18A and S18B Fig, S6 and S7 Data). Further bead aggregation assays focused on determining the minimum adhesive units for heterophilic adhesion. We carried out heterophilic binding assays using PCDH24 EC1-4Fc mixed with truncated fragments of CDHR5 and vice versa for both human and mouse proteins (Fig 7C–7F and 7I–7K, S18C–S18F Fig, S6 and S7 Data). The minimal adhesive units for heterophilic binding when using human proteins were PCDH24 EC1-2Fc and CDHR5 EC1Fc, whereas mouse proteins required PCDH24 EC1-2Fc and CDHR5 EC1-2Fc.

Additional experiments using only the PCDH24 and CDHR5 EC repeats required to mediate heterophilic adhesiveness for both human and mouse proteins confirmed the results obtained with the truncation series (Fig 7G and 7K, S18G and S18I Fig, S11 Data). Binding assays in the presence of 2 mM EDTA showed a lack of bead aggregation indicating that heterophilic adhesion mediated by PCDH24 and CDHR5 is calcium dependent, as expected (Fig 7H and 7K, S18H and S18J Fig, S12 Data). In addition, surface plasmon resonance (SPR) experiments revealed that the minimal adhesive units for the mouse proteins, PCDH24 EC1-2Fc and CDHR5 EC1-2, interact with a dissociation constant ($K_D$) of 16.0 ± 0.05 μM (S19 Fig, S13 Data), weaker than the heterophilic mouse CDH23 EC1-2 and PCDH15 EC1-2 complex ($K_D$ approximately 1 to 4 μM) [36,60,61] but comparable to measurement of homophilic binding for classical cadherins ($K_D$ approximately 20 to 150 μM) [62]. The low sequence similarity of PCDH24 and CDHR5 across different species may explain the differences in their binding mechanisms, as demonstrated by the distinct minimal adhesive units found for the human and mouse heterophilic complexes. Remarkably, while homophilic adhesive behavior is not conserved between human and mouse, the heterophilic bond is robust with slight differences in underlying molecular mechanisms.

## Discussion

Our sequence analyses, structures, and bead aggregation assays suggest that there are various modes of *trans* homophilic and heterophilic interactions involving the extracellular domains of PCDH24 and CDHR5 (Fig 8A). We have confirmed that human PCDH24 can form homophilic bonds [9] and found 3 potentially distinct modes of *trans* adhesion in bead aggregations assays. The physiological PCDH24 *trans* homophilic bond might be initiated by weak interactions involving EC1 and EC2 repeats (small bead aggregates), with further interdigitation for stronger interactions (larger bead aggregates) mediated by at least EC1 to EC3 repeats. A bona fide interface observed in the *hs* PCDH24 EC1-2 II structure, validated through mutagenesis, suggests specifically how one of the human homophilic PCDH24 complexes forms (Fig 8B). Since critical residues involved in this interface are at the base of EC1, it is possible that this complex mediates all modes of adhesion and that the strength of the bond and aggregate size is simply modulated by the presence of additional EC repeats.

Intriguingly, our bead aggregation assays indicate that mouse PCDH24 does not mediate *trans* homophilic adhesion, highlighting a species-dependent behavior that is consistent with poor sequence and structural conservation for the mouse and human proteins. The tyrosine residue in the human protein (p.Y71), important for *trans* homophilic adhesion as suggested

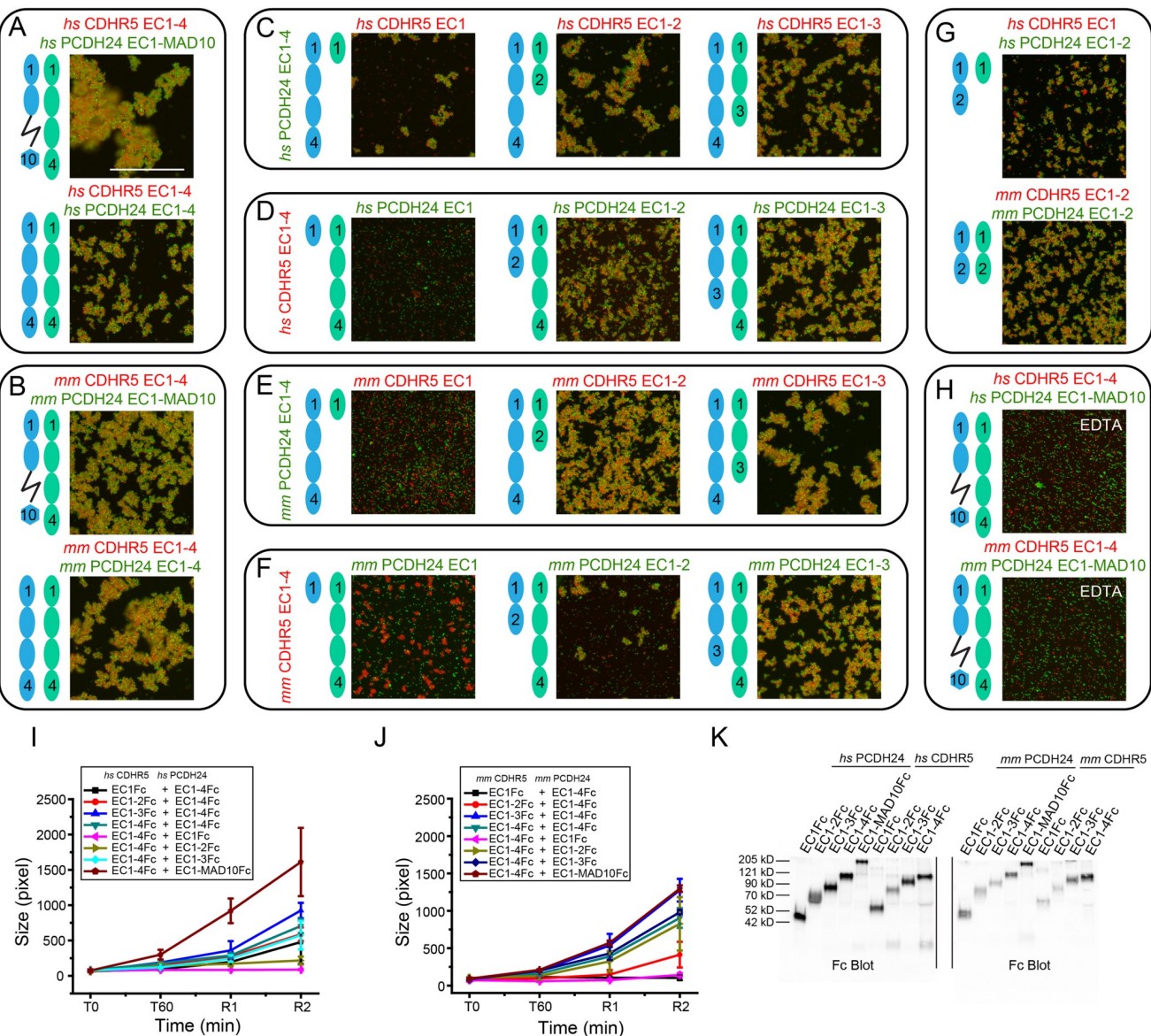

**Fig 7. Heterophilic binding assays of *hs* and *mm* PCDH24 and CDHR5.** (**A, B**) Images from binding assays of Fc-tagged *hs* PCDH24 (full-length extracellular domain and EC1–4Fc) mixed with the Fc-tagged full-length *hs* CDHR5 cadherin extracellular domain (A) and Fc-tagged *mm* PCDH24 (full-length extracellular domain and EC1-4Fc) mixed with Fc-tagged full-length *mm* CDHR5 cadherin extracellular domain (B). Green fluorescent Protein A beads are coated with Fc-tagged PCDH24 fragments and red fluorescent Protein A beads are coated with Fc-tagged CDHR5 in all panels. All images are details of complete views shown in S18 Fig. Images show bead aggregation observed after 60 min followed by rocking for 2 min in the presence of 2 mM CaCl₂. (**C-F**) Images from binding assays of truncations of Fc-tagged *hs* CDHR5 mixed with *hs* PCDH24 EC1-4Fc (C), truncations of Fc-tagged *hs* PCDH24 mixed with *hs* CDHR5 EC1-4Fc (D), truncations of Fc-tagged *mm* CDHR5 mixed with *mm* PCDH24 EC1-4Fc (E), and truncations of Fc-tagged *mm* PCDH24 mixed with *mm* CDHR5 EC1-4Fc (F). Images show bead aggregation observed after 60 min followed by rocking for 2 min in the presence of 2 mM CaCl₂. (**G**) Images from binding assays of minimum EC repeats required for heterophilic adhesion of *hs* and *mm* PCDH24 and CDHR5. The minimum units for heterophilic adhesion for the human proteins are CDHR5 EC1Fc and PCDH24 EC1-2Fc. The minimum units for heterophilic adhesion for the mouse proteins are CDHR5 EC1-2Fc and PCDH24 EC1-2Fc. Images show bead aggregation observed after 60 min followed by rocking for 2 min in the presence of 2 mM CaCl₂. (**H**) Protein A beads coated with full-length *hs* and *mm* PCDH24 and CDHR5 cadherin extracellular domains (including MAD10 for PCDH24 and without the MLD for CDHR5) in the presence of 2 mM EDTA shown as in (A, B). Bar– 500 μm. (**I**) Aggregate size (S6 Data) for Fc-tagged full-length and truncated constructs of *hs* PCDH24 and *hs* CDHR5 cadherin extracellular domains (including MAD10 for PCDH24 when indicated and without the MLD for CDHR5) at the start of the experiment (T0), after 60 min (T60) followed by rocking for 1 min (R1) and 2 min (R2). (**J**) Aggregate size (S7 Data) for Fc-tagged full-length and truncated constructs of *mm* PCDH24 and *mm* CDHR5 as in (I). Error bars in I and J are standard error of the mean (*n* indicated in S4 Table). (**K**) Western blot shows expression and secretion of Fc-tagged full-length and truncated versions of *hs* and *mm* PCDH24 and CDHR5 cadherin extracellular domains (including MAD10 for PCDH24 when indicated and without the MLD for CDHR5; S1 Raw Images). CDHR5, cadherin-related family member 5; EC, extracellular cadherin; MLD, mucin-like domain; PCDH24, protocadherin-24.

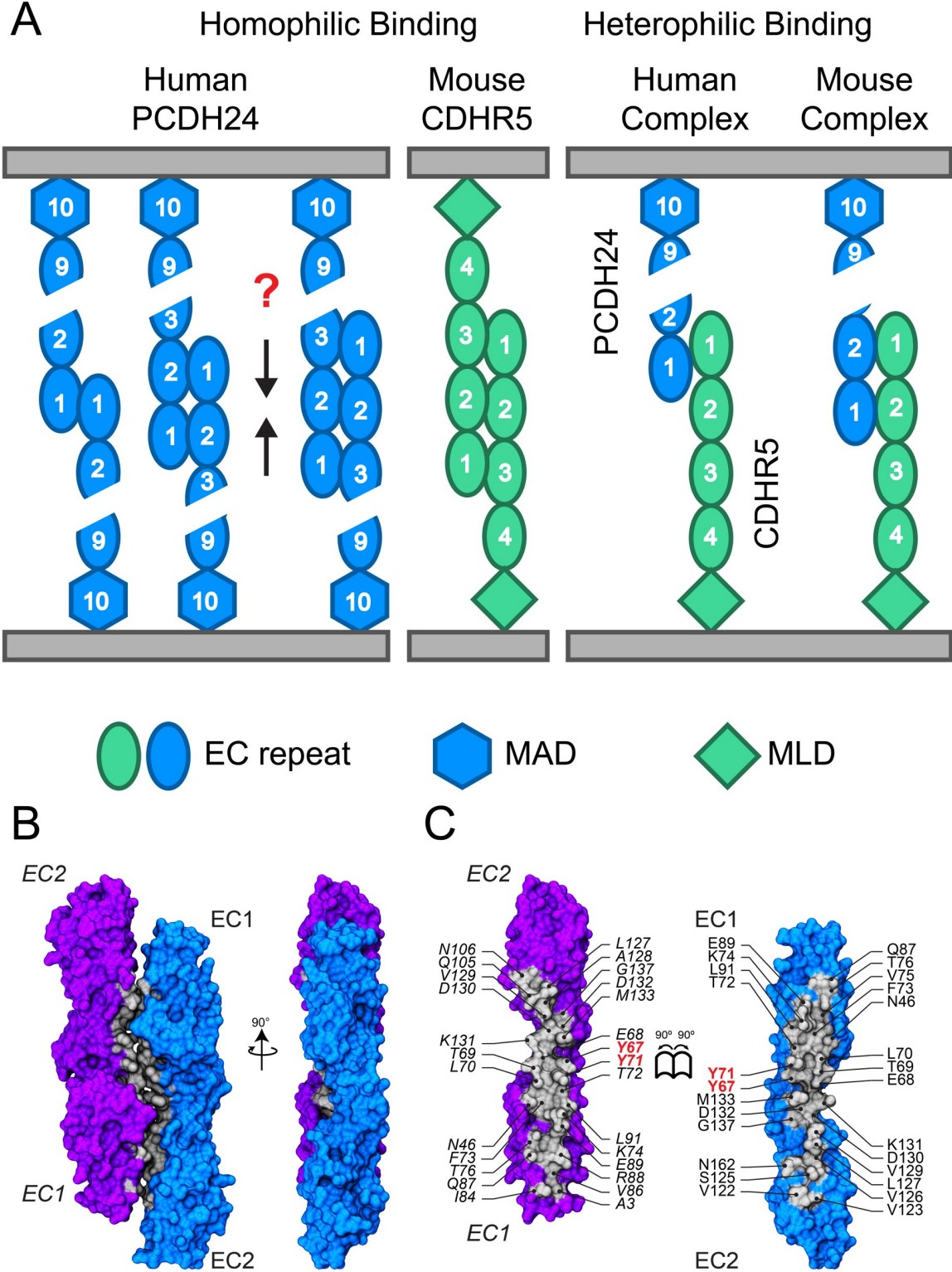

**Fig 8. Binding modes for PCDH24 and CDHR5 extracellular domains.** (**A**) Schematics of homophilic and heterophilic adhesion mediated by human and mouse PCDH24 and CDHR5. Homophilic adhesion that results in small bead aggregates can be mediated by *hs* PCDH24 EC1 and EC1-2, while homophilic adhesion that results in large bead aggregates requires *hs* PCDH24 EC1-3. Homophilic adhesion mediated by *mm* CDHR5 requires EC1-3. Heterophilic adhesion is mediated by PCDH24 EC1-2 and CDHR5 EC1 when using human proteins and by PCDH24 EC1-2 and CDHR5 EC1-2 when using mouse proteins. (**B**) Crystallographic antiparallel interface in *hs*

PCDH24 EC1-2 II. Molecular surface representation of 2 *hs* PCDH24 EC1-2 monomers in the crystal with the interaction interface formed by overlapped antiparallel EC1-2 repeats. Two perpendicular views are shown. (**C**) Interaction surface exposed with interfacing residues listed and shown in silver. CDHR5, cadherin-related family member 5; EC, extracellular cadherin; MAD, membrane adjacent domain; MLD, mucin-like domain; PCDH24, protocadherin-24.

by our structure and bead aggregation assays, is not conserved, and it is switched to a smaller proline (p.P71) in the mouse protein. This may partially explain the lack of homophilic adhesion mediated by mouse PCDH24. In contrast, experiments with CDHR5 reveal that the mouse protein can mediate *trans* homophilic adhesion, but the human protein cannot. These results are consistent with L929 cell aggregation assays that showed homophilic adhesion mediated by the rat CDHR5 protein [31,63]. The physiological relevance of the *trans* homophilic bonds for human PCDH24 and mouse CDHR5 remains to be determined. Mice lacking PCDH24 expression in the intestinal track are viable, but intestinal tissue exhibits various structural defects leading to functional impairment of intestine function and loss of body weight [8]. Intermicrovillar links formed by homophilic CDHR5 complexes might be partially compensating for the lack of heterophilic intermicrovillar links in these mice.

Importantly, adhesion mediated by PCDH24 and CDHR5 *trans* heterophilic bonds is species independent in our bead aggregation assays with mouse and human proteins. These results also demonstrate that lack of homophilic adhesion for mouse PCDH24 and for human CDHR5 is not caused by issues with bead coating or with protein misfolding and degradation in our assays, as similar, if not identical, conditions are used in bead aggregation experiments. Details of the *trans* heterophilic interaction between PCDH24 and CDHR5, however, might differ from species to species, as we found that the minimal adhesive units are PCDH24 EC1-2Fc/CDHR5 EC1Fc for human proteins and PCDH24 EC1-2Fc/CDHR5 EC1-2Fc for mouse proteins, with an affinity for the mouse PCDH24 EC1-2Fc and CDHR5 EC1-2 complex ($K_D$ approximately 16 μM) that is similar to affinities measured for other cadherin complexes [36,62]. Experimentally validated structural models for the human and mouse *trans* heterophilic complexes are still missing, but we can speculate that their architecture might be similar to what we observe for the human *trans* homophilic PCDH24 complex (Fig 8B), in which case an overlap of EC1-2 repeats may occur (S20 Fig). A simple structural alignment using the *hs* PCDH24 EC1-2 II structure and its in vitro validated *trans* interface as a template for heterophilic complexes suggests that some conformational rearrangements, perhaps through changes in inter-repeat angles, would be needed to avoid mild steric clashes. Nevertheless, residues important for bead aggregation mediated by the heterophilic complexes, such as p.R84 in *hs* CDHR5 EC1 and p.R82 and p.E84 in *mm* CDHR5, are at the interfaces of these models, which might represent good starting configurations for the *trans* heterophilic complexes (S20 Fig).

Our structures of the human PCDH24 EC1-2, mouse PCDH24 EC1-3, and human CDHR5 EC1-2 protein fragments confirm the expected structural similarities between the PCDH24 and CDH23 tips and between the PCDH15 and CDHR5 tips. These structures also support binding mechanisms that are distinct from those used by classical cadherins, clustered protocadherins, and nonclustered δ protocadherins [43,45,46,48–52,64,65]. While crystal contacts in our structures suggest possible interfaces that might facilitate the modes of adhesion discovered through bead aggregation assays, only one of the crystallographic interfaces (*trans* homophilic in *hs* PCDH24 EC1-2 II; Fig 8B and 8C) withstood in vitro validation. It is important to note that our in vitro bead aggregation assays, performed with beads that are saturated with coating Fc-tagged proteins, are semiquantitative, and that both protein quantity and bond affinity will control adhesion in vivo [66]. While our data suggest that homophilic *trans* interactions are not required for heterophilic aggregation, quantification of the relative strengths

and affinities of all homophilic and heterophilic bonds is still missing. Further work will be required to elucidate and validate structural models of *trans* homophilic and heterophilic complexes, to determine whether *cis* interactions occur and are relevant for *trans* interactions, to clarify the structural role played by glycosylation in vivo [67], and to determine which complexes form intermicrovillar links in enterocytes.

Intermicrovillar links observed in human CACO-2$_{BBE}$ cells and in native mouse tissue are 46.8 ± 8.9 nm and 49.9 ± 8.8 nm in length, respectively [9]. Interestingly, lengths range from 30 to 80 nm for both human and mouse links, perhaps reflecting the challenges of measuring link sizes from two-dimensional electron microscopy images of cells, with some additional uncertainty coming from difficult-to-define link edges. Nevertheless, this range of lengths hints at multiple possibilities for the architecture of the complex that are consistent with the modes of adhesion we found using bead aggregation assays. Links made of 2 PCDH24 molecules interacting in *trans* and tip-to-tip could reach lengths of up to approximately 90 nm (9 EC repeats + MAD10 per molecule; 4.5 nm long each), while links made of the shortest CDHR5 isoforms with just 4 EC repeats interacting tip-to-tip could be 36 nm in length or even shorter if there is any overlap at the junction or bending at any of the EC linkers. Heterophilic links formed by a tip-to-tip *trans* interaction of the full-length PCDH24 ectodomain and the shortest CDHR5 isoform should be approximately 63 nm or shorter. The average value observed ex vivo and in vivo (approximately 48 nm) indicates that some overlap at the junction may exist and that most of the links are likely formed by the heterophilic complex. The minimal adhesive units we found for mouse and human PCDH24 and CDHR5 homophilic and heterophilic *trans* complexes are consistent with these estimates for length ranges, assuming that up to 3 EC repeats might be overlapping to form robust bonds that lead to bead aggregation in vitro and adhesion in vivo.

The functional role of PCDH24 and CDHR5 in nonintestinal tissues and cells is unclear. These proteins have also been found in the liver and kidney [31,68] and have been implicated in contact inhibition of cell proliferation [68,69], morphogenesis [31,37,67], colorectal carcinogenesis [70,71], and gallstone disease [72]. Exploring how different organs in different species express and use PCDH24 and CDHR5 proteins, which have poorly conserved sequences, might provide with a unique opportunity to reveal how evolutionary events structurally encode adhesive and signaling functions in various physiological contexts [73]. It is tempting to speculate that differences in gut proteases' specificity across species, as well as diverse diets and gut microbiota, put divergent selective pressures driving the molecular evolution of intermicrovillar links. Similarly, other selective pressures relevant to specific organs might be driving the evolution of heterophilic and homophilic PCDH24 and CDHR5 adhesive complexes.

Interestingly, both brush border enterocytes and inner-ear sensory receptor cells use heterophilic cadherin complexes in actin-based structures relevant for their specialized functions. These heterophilic cadherin complexes link membranes within cells, as opposed to mostly homophilic cadherin complexes that mediate adhesion between membranes of adjacent cells. Heterophilic complexes, with a pair of distinct cytoplasmic domains, might be more functionally versatile in signaling, as the number and type of cytoplasmic partners can be independently diversified for each member of the complex. The strength and dynamics of cadherin heterophilic bonds, which are present in systems subjected to mechanical stimuli, might also differ from and provide specific advantages over weak homophilic cadherin complexes that cluster to provide strong adhesiveness. Whether PCDH24 and CDHR5 engage in *cis* interactions that can lead to more mechanically robust intermicrovillar links, as observed for tip links [28,40,74], or to the formation of larger adhesive complexes in nonmicrovillar membranes, as observed for classical cadherins and clustered protocadherins [43,53,64], remains to be explored.

## Methods

### Cloning of PCDH24 and CDHR5 fragments

DNA encoding for *hs* PCDH24 EC1-2, *mm* PCDH24 EC1-3, and *hs* CDHR5 EC1-2, without signal peptides, was subcloned into *NdeI* and *XhoI* sites of the pET21a vector including a fused C-terminal hexa-histidine (His) tag. DNA encoding for full-length extracellular domains and C-terminal truncations of *hs* and *mm* PCDH24 as well as of *hs* and *mm* CDHR5, including signal peptides, were cloned into modified CMV vectors (Jontes laboratory, OSU) including fused C-terminal His or Fc tags. Truncations were made at DXNDN or similar ends of each EC repeat (S1 and S2 Figs), and cloning sites were *XhoI* and *KpnI* (*hs* PCDH24, *mm* PCDH24), *NheI* and *XhoI* (*hs* CDHR5), and *XhoI* and *BamHI* (*mm* CDHR5). The sequence for the linker between the PCDH24 protein fragments and their tags in CMV vectors is TVPRARDPPV (with 2 additional GG residues before the His tag but not the Fc tag), while the sequence for the linker between the *hs* CDHR5 protein fragments and their tags in CMV vectors is LELKLRILQSTVPRARDPPV(GG), and between the *mm* CDHR5 protein fragments and their tags in CMV vectors is TDPPV(GG). Mutations were generated using the Quik-Change Lightning kit (Agilent). All DNA constructs were sequence verified. All protein products mentioned in the text have a His tag unless otherwise explicitly stated.

### Bacterial expression and purification of PCDH24 and CDHR5 fragments

DNA constructs based on the pET21a vector were used to transform BL21(DE3) pLysS cells (Stratagene). The PCDH24 transformants were cultured in TB and induced at $OD_{600}$ approximately 0.6 with 1 mM IPTG at 37 ˚C overnight. The *hs* CDHR5 EC1-2 transformant was cultured in TB and induced at $OD_{600}$ approximately 0.6 with 1 mM IPTG at 30 ˚C overnight. Cells were lysed by sonication in denaturing buffer (20 mM Tris-HCl [pH 7.5], 6 M guanidine hydrochloride [GuHCl], 10 mM $CaCl_2$, and 20 mM imidazole). The cleared lysates were loaded onto Ni-Sepharose beads (GE Healthcare) and nutated for approximately 1 h. Beads were washed twice with denaturing buffer, and his-tagged proteins were eluted with the same buffer supplemented with 500 mM imidazole. After purification, 2 mM DTT was added to the protein (<1 mg/ml) before refolding. The *hs* PCDH24 EC1-2 protein was refolded in 6 dialysis steps using a buffer with 20 mM Tris-HCl [pH 8.0], 5 mM $CaCl_2$, and decreasing amounts of GuHCl (6 M, 3 M, 2 M, 1 M, 0.5 M, and 0 M). In steps 4, 5, and 6, the refolding buffer included 400 mM arginine. Steps 3 through 6 also included 150 mM NaCl. The *mm* PCDH24 EC1-3 and *hs* CDHR5 EC1-2 proteins were refolded overnight using a buffer with 400 mM Arginine, 20 mM Tris-HCl [pH 8.0], 150 mM NaCl, and 5 mM $CaCl_2$. All dialysis steps were performed using MWCO 2000 membranes and used refolding buffers for 12 h at 4 ˚C. Refolded protein was further purified on a Superdex200 16/600 column (GE Healthcare) in 20 mM Tris-HCl [pH 8.0], 150 mM NaCl, and 2 mM $CaCl_2$ and concentrated by ultrafiltration to approximately 5 mg/ml for crystallization (Vivaspin 10 KDa).

### Mammalian expression and purification of PCDH24 and CDHR5 fragments

The Expi293 expression system (Gibco) was used to produce *hs* PCDH24 EC1-2 and CDHR5 EC1-2 for crystallization screens, as well as *mm* PCDH24 EC1-2Fc and *mm* CDHR5 EC1-2 for SPR experiments. Suspension-cultured Expi293F cells were transfected using the ExpiFecta-mine 293 transfection kit (Gibco) following the standard protocol provided for all our DNA constructs, except for *mm* CDHR5 EC1-2, for which we used twice the amount of recommended DNA. A glycosylation inhibitor (1 mM kifunensine) was added to cells transfected

with *hs* PCDH24 EC1-2 and CDHR5 EC1-2. Transfection enhancers were added 1 day after transfection, and media was collected 2 to 3 days thereafter. Media for the His-tagged proteins was dialyzed in a buffer containing 20 mM Tris [pH 7.5], 150 mM KCl, 50 mM NaCl, and 10 mM $CaCl_2$ to remove EDTA. One day after dialysis, the media was concentrated (Amicon 10 kDa) and incubated with Ni-Sepharose beads (GE Healthcare) with nutation for approximately 1 h. Beads were washed 3 times with buffer containing 20 mM Tris HCl [pH 8.0], 300 mM NaCl, 10 mM $CaCl_2$, and 20 mM imidazole. The His-tagged proteins were eluted with the same buffer supplemented with 500 mM imidazole. Media for the Fc-tagged proteins was concentrated (Amicon 10 kDa) and incubated with nutation with MabSelect PrismA protein A resin (Cytiva) for 1 h. The resin was washed 3 times with 20 mM Tris HCl [pH 7.5], 300 mM NaCl, and 10 mM $CaCl_2$. The Fc-tagged protein was eluted with 0.1 M glycine [pH 3.0], 200 mM NaCl, and 10 mM $CaCl_2$ and immediately neutralized with 1 M Tris HCl [pH 8.0], 200 mM NaCl, 10 mM $CaCl_2$, and 0.5 mM TCEP using a 10:1 ratio of protein sample to neutralizing buffer. All proteins were further purified on a Superdex200 10/300 column (GE Healthcare) in 20 mM Tris-HCl [pH 8.0], 150 mM KCl, 50 mM NaCl, and 2 mM $CaCl_2$. The *hs* PCDH24 EC1-2 and CDHR5 EC1-2 proteins were subsequently copurified using the same column. The *mm* PCDH24 EC1-2Fc protein was purified in buffer supplemented with 0.5 mM TCEP. Eluted protein samples for crystallization were concentrated by ultrafiltration (Vivaspin 10 kDa) to approximately 1 mg/mL for crystallization screens. Concentration of the eluted protein samples for SPR experiments was determined using a Nanodrop 2000C instrument (Thermo Scientific). These samples were subsequently diluted, as described below.

## Crystallization, data collection, and structure determination

Crystals for bacterially produced proteins were grown by vapor diffusion at 4 ˚C by mixing 1 μL of protein sample and 0.5 μL of reservoir solution including (0.2 M MgCl, 0.1 M HEPES [pH 7.3], 10% PEG 4000) for *hs* PCDH24 EC1-2 (form I structure), (3.0 M LiCl, 0.1 M HEPES [pH 7.5]) for *mm* PCDH24 EC1-3, and (0.1 M Na Acetate [pH 4.4], 0.7 M $CaCl_2$) for *hs* CDHR5 EC1-2. Crystals for mammalian-produced *hs* PCDH24 EC1-2 (form II structure) were grown as described above but with a mixture of 0.5 μL of protein sample that included mammalian-produced *hs* CDHR5 EC1-2 (not present in the structure) and 0.5 μL of reservoir solution (0.2 M NaI, 20% PEG 3350). Crystals were cryo-protected in reservoir solution plus a cryo protectant (25% PEG 400 for PCDH24 crystals and 25% glycerol for *hs* CDHR5 EC1-2 crystals) and cryo-cooled in liquid $N_2$. X-ray diffraction datasets were collected as indicated in Table 1 and processed with HKL2000 [75].

All structures were solved by molecular replacement using PHASER [76]. The first 2 N-terminal repeats of CDH23 EC1-2 (PDB: 4AQE) [36] were used as an initial model for the *mm* PCDH24 EC1-3 structure. The *mm* PCDH24 EC3 repeat was built using BUCCANEER [77,78]. The *hs* PCDH24 EC1-2 I structure was determined by using the first 2 N-terminal EC repeats of the *mm* PCDH24 EC1-3 structure (PDB: 5CYX). The *hs* PCDH24 EC1-2 II structure was solved using the *hs* PCDH24 EC1-2 I structure. For the *hs* CDHR5 EC1-2 structure, we used a RaptorX [79] model created using CDH23 EC22-24 (PDB: 5UZ8) [39]. Model building was done with COOT [80], and refinement was performed with REFMAC5 [81]. Restrained TLS refinement was used for the *hs* PCDH24 EC1-2 I and *mm* PCDH24 EC1-3 structures. Data collection and refinement statistics are provided in Table 1.

## Bead aggregation assays

Bead aggregation assays were performed as previously described [9,45,46,82,83]. The PCDH24Fc and CDHR5Fc fusion constructs were transfected into HEK293T cells via

calcium-phosphate transfection [84–86]. A solution of 10 µg of DNA and 250 mM $CaCl_2$ was added dropwise to 2X HBS with mild vortexing. The solution was immediately added dropwise to a 100-mm dish of HEK293T cells cultured in DMEM supplemented with FBS, L-glutamine, and PenStrep. For each construct, 2 plates of cells were transfected. The following day, cells were rinsed twice with 1X PBS and allowed to grow in serum-free media for 2 days prior to collection. The collected media contained the secreted Fc fusion proteins and was concentrated (Amicon 10 kDa and 30 kDa) to a volume of 500 µL before being incubated with 1.5 µL of Protein G Dynabeads (Invitrogen). The beads were incubated with rotation for 2 h at 4 ˚C. Incubated beads were washed with binding buffer (50 mM Tris [pH 7.5], 100 mM NaCl, 10 mM KCl, and 0.2% BSA) and split into 2 tubes, either with 2 mM $CaCl_2$ or 2 mM EDTA. Beads were then allowed to aggregate on a glass depression slide in a humidified chamber for 1 h, then rocked for 1 to 2 min at 8 oscillations/min. Images were taken at various time points using a Nikon Eclipse Ti microscope with a 10X objective as detailed in S4 Table. Sizes of bead aggregates were quantified with ImageJ as described previously [82,83]. Images were thresholded to exclude background, and the area of the detected particles was measured in pixels. Average aggregate size (±SEM) was plotted in Origin Pro. Number of independent biological replicates are listed in S4 Table.

### Fluorescent bead aggregation assays

Fluorescently labeled Protein A Dynabeads were produced for heterophilic bead aggregation assays. Protein A (5 mg; Thermo Fisher Pierce) was resuspended in 1X PBS to obtain a final concentration of 2 mg/mL. Using the Thermo Fisher Dylight Antibody Labeling kits, 1 mg of Protein A was labeled with Dylight 488 to produce 0.5 moles of dye/1 mole of Protein A. Labeled Protein A (200 µg) was then conjugated to Dynabeads M-280 Tosylactivated (Invitrogen) following the provided protocol to produce 10 mg of beads at a final bead concentration of 20 mg/mL. A similar protocol was used to obtain Dylight 594 conjugated Protein A Dynabeads. Conjugation to the beads was confirmed by observing the beads in depression slides under 10X magnification with the GFP filter cube (Nikon) for the Dylight 488 fluorophore and the Texas Red filter cube (Nikon) for the Dylight 594 fluorophore. Beads of both colors were then mixed on a depression slide, and no overlap of fluorescence was observed. Further confirmation that the beads were functional was done by performing a bead aggregation assay with a positive control (*hs* PCDH24 EC1-4Fc).

Fluorescent bead aggregation assays were performed as done with nonfluorescent beads but with noted exceptions. Concentrated media was added to 2.2 µL of fluorescently labeled beads to achieve the same concentration as the Protein G Dynabeads on the slides. Dylight 488 conjugated beads were used for all PCDH24Fc fusions, and Dylight 594 conjugated beads were used for all CDHR5Fc fusions. Prior to addition of beads to the $CaCl_2$ or EDTA tubes, 160 µL of Dylight 488 beads containing the PCDH24Fc constructs were mixed with 160 µL of Dylight 594 beads containing CDHR5Fc constructs, then 150 µL of the mixture was added to the $CaCl_2$ or EDTA tubes.

### Western blots

Western blots were performed for Fc-tagged proteins to confirm expression and secretion of the protein. Protein samples were mixed with 4X SDS with β-mercaptoethanol, boiled for 5 min, and loaded for SDS-PAGE (BioRad) experiments. Proteins were electroblotted onto PVDF membrane (GE Healthcare), blocked with 5% nonfat milk in TBS with 0.1% Tween-20 for 1 h, and incubated overnight at 4 ˚C with goat anti-human IgG (1:200; Jackson ImmunoResearch Laboratories, Catalog-109-025-003, Lot-117103, 135223). After washing with 1X TBS,

the membranes were incubated with mouse anti-goat HRP-conjugated secondary antibody (1:5,000; Santa Cruz Biotechnology, Catalog-sc-2354, Lot-A2017, J1718) for 1 h at room temperature. Membranes were washed in 1X TBS and developed using the ECL Select western blot detection kit (GE Healthcare) for chemiluminescent detection (Omega Lum G). Western blots were performed on samples obtained immediately after media collection and concentration, as well as on protein samples obtained from the Protein G/A beads following aggregation assays.

## Surface plasmon resonance experiments

The interaction between *mm* PCDH24 EC1-2Fc and *mm* CDHR5 EC1-2 was analyzed using an OpenSPR 2-Channel instrument (Nicoya) and a Protein A sensor chip (Nicoya). This sensor chip was prepared following the provided protocol in PBS buffer, up through the blocking step. The sensor was then equilibrated in SPR running buffer containing 20 mM Tris HCl [pH 7.5], 150 mM NaCl, 50 mM KCl, 2 mM $CaCl_2$, 0.5 mM TCEP, and 1% BSA for 1 h prior to the experiment. Following equilibration, *mm* PCDH24 EC1-2Fc (0.5 mg/mL) was immobilized on the sensor chip with sample running through channel 2 until a surface density equivalent to >800 resonance units (RUs) was achieved. Data were collected by injecting *mm* CDHR5 EC1-2 at concentrations ranging from 2 μM to 23.5 μM (diluted from high-concentration stocks) and buffer matched to the SPR running buffer. All steps used a flow rate of 20 μL/min. Each injection was started after the signal achieved baseline (approximately 10 min). A buffer blank was run by injecting SPR running buffer after 2 injections of CDHR5. Binding curves were analyzed and fit using the TraceDrawer software (Nicoya) to obtain $K_D$, $k_{on}$, and $k_{off}$ using a kinetic evaluation assuming a 1:1 binding interaction. In rare instances, the signal for an injection appeared to be higher or lower than expected as compared to previous or subsequent injections. In such cases, these injections were excluded from data analysis. The values reported for the $K_D$, $k_{on}$, and $k_{off}$ were calculated from at least 7 concentrations included in the theoretical fit to the data.

## Sequence alignments

MUSCLE [87] was used to do alignments of individual EC repeat sequences from PCDH24, CDHR5, CDH23, and PCDH15. These alignments included sequences from 13 species that covered the complete extracellular domains of these proteins and that were evolutionarily diverse (S2 Table). Sequence conservation per EC repeat in percent identity was obtained from the MUSCLE alignments using Geneious [88] (S1 Table). All EC repeats of both human and mouse PCDH24 (S5 Fig) and CDHR5 (S15 Fig) were also aligned to one another as described above. Sequence conservation was visualized in alignments using JalView [89] and colored by percent identity [90] with a conservation cutoff of 40%. MUSCLE was also used to do a separate set of alignments that included sequences of the N-terminal repeats (EC1-3) of PCDH24, CDHR5, CDH23, PCDH15, CDH1, and CDH2 (S3 Table). The Sequence Identity and Similarity Server (SIAS) was used to determine identity between pairs of different species for each protein (Fig 1B). Average percent identity across species and EC repeats and for pairwise comparisons were computed from all reported values (Fig 1B and S1 Table).

A broader conservation analysis of PCDH24 EC1-3 and CDHR5 EC1-2 was done in Consurf [91–95] using additional sequences to include a larger variety of species (S5 Table for PCDH24 and S6 Table for CDHR5). The values obtained from Consurf (1 = lowest conservation, 9 = highest conservation) were matched to the corresponding residues in the structures and used to make a conservation heat map. N-glycosylation and O-glycosylation sites for the

extracellular domains of PCDH24 and CDHR5 were predicted using NetNGlyc 1.0 and NetOgly 4.0 [96] and highlighted in the multiple sequence alignment.

## Simulated systems

The psfgen, solvate, and autoionize VMD [97] plug-ins were used to build all systems (S7 Table), adding hydrogen atoms to the protein structure and crystallographic water molecules. For the *mm* PCDH24 EC1-3 systems, the missing loop in EC1 (residues 32 to 37) was built in COOT [80] using the human PCDH24 EC1 structure as a template. Nonnative N-terminal residues were removed. Residues D, E, K, and R were considered to be charged, and neutral histidine protonation states were determined on favorable hydrogen bond formation. The proteins were placed in additional water and randomly placed ions to solvate and ionize the systems at 150 mM NaCl. For SMD simulations, protein molecules were aligned to the *x* axis. Additional information for systems is presented in (S7 Table).

## Simulations parameters

Equilibrium and SMD simulations were performed using NAMD 2.11/2.12 [98], the CHARMM36 force field for proteins with the CMAP correction, and the TIP3P model for water [99]. Van der Waals interactions were computed with a cutoff at 12 Å and using a switching function starting at 10 Å. Computation of long-range electrostatic forces was done with the Particle Mesh Ewald method with a cutoff of 12 Å and a grid point density of $>1$ Å$^{-3}$. A 2-fs uniform integration time was used with SHAKE. When indicated, constant temperature ($T = 300$ K) was enforced using Langevin dynamics with a damping coefficient of $\gamma = 0.1$ ps$^{-1}$ unless otherwise noted. The hybrid Nosé-Hoover Langevin piston method was used to maintain constant pressure ($NpT$) at 1 atm with a 200-fs decay period and a 100-fs damping time constant. In simulation Sim2a (S7 Table), the *mm* PCDH24 EC1-3 protein had harmonic constraints ($k = 1$ kcal mol$^{-1}$ Å$^{-2}$) applied to the C$_\alpha$ atoms of residues S110, V123, and L182 in EC2 to avoid rotations leading to interactions between periodic images. Constant velocity stretching simulations (Sim2b-Sim2d, Sim3b-Sim3d, and Sim4b-Sim4d in S7 Table) were carried out using the SMD method and the NAMD Tcl interface [100–103]. Virtual springs ($k_s = 1$ kcal mol$^{-1}$ Å$^{-2}$) were attached to the center of mass of EC1 and EC3 for unfolding simulations (Sim2b-Sim2d) and to Cα atoms of C-terminal ends for unbinding simulations (Sim3b-Sim3d and Sim4b-Sim4d). The free ends of the stretching springs were moved in opposite directions, away from the protein(s), at a constant velocity. Applied forces were computed using the extension of the virtual springs. Maximum force peaks and their averages were computed from 50-ps running averages.

Data for bead aggregation experiments deposited in the Dryad repository: https://doi.org/10.5061/dryad.w0vt4b8sh [104].

## Accession numbers

Coordinates for *hs* PCDH24 EC1-2 I, *hs* PCDH24 EC1-2 II, *mm* PCDH24 EC1-3, and *hs* CDHR5 EC1-2 have been deposited in the Protein Data Bank with entry codes 5CZR, 7N86, 5CYX, and 6OAE, respectively.

## Supporting information

**S1 Fig. Sequence alignments of individual EC repeats of PCDH24.** Multiple sequence alignments comparing each EC repeat and MAD10 of PCDH24 from 13 different species. Each alignment is colored by percent identity, with white being the lowest percent identity and dark

blue being the highest. Sites of N-linked (blue in human, green in mouse) and O-linked (purple in human, red in mouse) glycosylation are denoted by a colored circle. An asterisk (*) indicates sites Y67 and Y71 mutated in binding assays. Secondary structure elements observed in the crystal structures of *hs* PCDH24 EC1-2 and *mm* PCDH24 EC1-3 are illustrated below the respective repeats. Residues at the antiparallel homophilic *hs* PCDH24 EC1-2 II interface (>30% buried surface area) are denoted by a light blue bar. Calcium-binding motifs are indicated above the sequences, which are numbered according to the human protein. Species are abbreviated as follows: *Homo sapiens* (*Hs*), *Mus musculus* (*Mm*), *Sus scrofa* (*Ss*), *Gallus gallus* (*Gg*), *Aptenodytes forsteri* (*Af*), *Parus major* (*Pm*), *Anolis carolinensis* (*Ac*), *Crocodylus porosus* (*Cp*), *Thamnophis elegans* (*Te*), *Danio rerio* (*Dr*), *Oryzias melastigma* (*Om*), *Mastacembelus armatus* (*Ma*), and *Xenopus tropicalis* (*Xt*). Species were chosen based on sequence availability, coverage of entire extracellular domain, and taxonomical diversity. Accession numbers and species can be found in S2 Table. EC, extracellular cadherin; PCDH24, protocadherin-24. (PDF)

**S2 Fig. Sequence alignments of individual EC repeats of CDHR5.** Multiple sequence alignments comparing each EC repeat of CDHR5 from 13 different species, shown as in S1 Fig. An asterisk (*) indicates site R84 mutated in binding assays. An arrow indicates the end of EC1-EC4 protein fragments used in binding assays. Secondary structure elements observed in the crystal structures of *hs* CDHR5 EC1-2 are illustrated below the respective repeats. Calcium-binding motifs are indicated above the sequences, which are numbered according to the human protein. Species are abbreviated as follows: *Homo sapiens* (*Hs*), *Mus musculus* (*Mm*), *Sus scrofa* (*Ss*), *Gallus gallus* (*Gg*), *Aptenodytes forsteri* (*Af*), *Parus major* (*Pm*), *Anolis carolinensis* (*Ac*), *Crocodylus porosus* (*Cp*), *Thamnophis elegans* (*Te*), *Danio rerio* (*Dr*), *Oryzias melastigma* (*Om*), *Mastacembelus armatus* (*Ma*), and *Xenopus tropicalis* (*Xt*). Species were chosen based on sequence availability and taxonomical diversity. Accession numbers and species can be found in S2 Table. CDHR5, cadherin-related family member 5; EC, extracellular cadherin. (PDF)

**S3 Fig. Comparison of PCDH24 and CDHR5 N-termini with other cadherins.** (**A**) Alignment of the processed N-terminal sequences of Cr-2 protocadherins *hs* PCDH24, *mm* PCDH24, *hs* CDH23, and *hs* PCDH21 and of 2 classical cadherins, *hs* CDH1 and *hs* CDH2. Calcium-binding sites are labeled and boxed in red in the sequence to show the additional calcium-binding sites in Cr-2 protocadherins not present in classical cadherins. Residue W2 involved in classical cadherin binding is boxed in pink. (**B**) Alignments of the processed sequences of EC1 for *hs* CDHR5, *mm* CDHR5, and *hs* PCDH15. Calcium-binding sites are labeled. The 2 cysteine residues that form a disulfide bond between β-strands A and F are highlighted by a red box. CDH1, Cadherin-1; CDH2, Cadherin-2; CDH23, Cadherin-23; CDHR5, cadherin-related family member 5; PCDH15, protocadherin-15; PCDH21, protocadherin-21; PCDH24, protocadherin-24. (PDF)

**S4 Fig. Stereo views of electron density maps (2Fo–Fc) for unique elements of *hs* PCDH24 EC1-2 I, *hs* PCDH24 EC1-2 II, *mm* PCDH24 EC1-3, and *hs* CDHR5 EC1-2.** (**A**) Extended F-G loop of EC2 in the *hs* PCDH24 EC1-2 I structure with the disulfide bond C187–C201. Electron density shown at 1.0 σ with a carve radius of 1.6 Å. This loop is also present in the *hs* PCDH24 EC1-2 II and *mm* PCDH24 EC1-3 structures. (**B**) Sugars (orange) from N-glycosylation at p.N9 in EC1 of the *hs* PCDH24 EC1-2 II structure. Electron density shown at 0.9 σ with a carve radius of 1.6 Å. (**C**) The atypical extended N-terminus seen in *mm* PCDH24 EC1. Electron density shown at 1.0 σ with a carve radius of 1.8 Å. (**D**) Disulfide bond between residue

C5 in β-strand A and residue C73 in β-strand F of the *hs* CDHR5 EC1-2 structure. Electron density shown at 1.0 σ with a carve radius of 1.6 Å. CDHR5, cadherin-related family member 5; PCDH24, protocadherin-24.
(PDF)

**S5 Fig. Sequence alignment of *hs* and *mm* PCDH24 EC repeats.** All 9 EC repeats for each species are aligned to each other (EC1 to EC9). Conserved calcium-binding motifs are labeled on top of the alignment. EC, extracellular cadherin; PCDH24, protocadherin-24.
(PDF)

**S6 Fig. Equilibrium and stretching simulations of *mm* PCDH24 EC1-3.** (**A**) Superposition of *mm* PCDH24 EC1-3 conformations taken every 5 ns from an approximately 99-ns-long trajectory (simulation Sim1; S7 Table). Repeat EC2 was used as a reference. Side, top, and bottom views are shown. Color indicates time step (red-white-blue). (**B**) Orientation projections illustrating the conformational freedom of the EC1-2 and EC2-3 linkers throughout equilibrium simulations. To quantify the conformational freedom of EC2 relative to EC1 (top) and of EC3 relative to EC2 (bottom), the longest principal axes of EC1 and EC2 were aligned to the *z* axis, and then the projections of the longest principal axes of EC2 (blue) and EC3 (red) in the *x*-*y* plane were plotted. The initial orientation of a control, CDH23 EC1-2 (PDB: 2WHV), is shown as a black dot [34]. The EC2-3 linker behavior is not dramatically different than the behavior observed for the EC1-2 linker, suggesting similar flexibility. (**C**) Trajectory snapshots during the slowest speed stretching simulation at 0.1 nm/ns for *mm* PCDH24 EC1-3 (Sim2d; S7 Table). Springs indicate position (center of mass) and direction of applied forces. Unfolding of the EC2-3 linker is observed first. End-to-end distances between the centers of mass of EC1 and EC3 are indicated for each snapshot. (**D**) Force versus end-to-end distance (S8 Data) for simulations of the *mm* PCDH24 EC1-3 monomer (Sim2b-Sim2d) at stretching speeds of 10 nm/ns (red), 1 nm/ns (blue), and 0.1 nm/ns (green). Dark and light colors indicate forces applied at opposite ends. Black arrowheads indicate time points illustrated in (C). (**E**) Detail of the *mm* PCDH24 EC2-3 linker at the force peak during the slowest stretching simulation at 0.1 nm/ns (Sim2d; S7 Table). Residues involved in two-step unbinding from calcium ions are shown. CDH23, Cadherin-23; PCDH24, protocadherin-24.
(PDF)

**S7 Fig. Structural comparison of PCDH24 and CDHR5 EC1 repeats with other cadherins.** (**A**) Detail of superposed EC1 N-termini from structures *hs* PCDH24 EC1-2 I (blue) and *mm* CDH23 EC1-2 (gray; PDB: 2WHV). Both have a calcium ion at the calcium-binding site 0, a feature of Cr-2 protocadherins. (**B**) Detail of superposed EC1 N-termini of *mm* PCDH24 (blue) and *hs* CDH1 (gray; PDB: 2O72). Both have N-termini protruding away from the monomer to interact with another neighboring EC1, as shown in (C) for *hs* CDH1 and (D) for *mm* PCDH24. (**C, D**) Surface representations of *hs* CDH1 and *mm* PCDH24, respectively. Highlighted in yellow is β-strand A with W2 from its binding partner. Residues 25, 26, and 27 are not shown to facilitate visualization of the tryptophan binding pocket. Highlighted in purple is the N-terminus of *mm* PCDH24 from a neighboring protomer in the crystal lattice (Fig 2D). (**E**) Detail of *hs* CDHR5 EC1 (green) and *mm* PCDH15 EC1 (purple) superposed. Disulfide bonds are shown in orange and yellow. PCDH15 has longer loops and a longer N-terminus, with a helix that interacts with CDH23. CDH1, Cadherin-1; CDH23, Cadherin-23; CDHR5, cadherin-related family member 5; PCDH15, protocadherin-15; PCDH24, protocadherin-24.
(PDF)

**S8 Fig. Crystal contacts in the *hs* PCDH24 EC1-2 I structure.** (**A-D**) Ribbon diagram of 2 monomers showing a crystal contact between chains A and D with an interface area of 979.2 Å$^2$. The arrangement is antiparallel, likely describing a possible *trans* interface (A). Crystal contacts in the entire asymmetric unit include 4 additional interfaces with areas of 173.6 Å$^2$ (chains A and B), 231.1 Å$^2$ (chains A and C), 255.3 Å$^2$ (chains D and B), and 307.2 Å$^2$ (chains D and C). Two additional interfaces between chains A and C (387.4 Å$^2$ and 507.3 Å$^2$) are shown in (C) and (D). Equivalent interfaces for the one shown in (A) between chains B and C, and those shown in (C, D) between chains D and B, have similar interface areas of 938.6 Å$^2$, 390.8 Å$^2$, and 515.1 Å$^2$, respectively, but are not shown. PCDH24, protocadherin-24.
(PDF)

**S9 Fig. Stretching simulations testing the strength of the *hs* PCDH24 EC1-2 I and II *trans* interfaces.** (**A**) Force versus end-to-end distance for simulations of the largest crystallographic *trans* interface (S8A Fig) observed for the *hs* PCDH24 EC1-2 I structure (simulations Sim3b-Sim3d; S7 Table, S9 Data). Forced unbinding simulations were carried out at stretching speeds of 10 nm/ns (red), 1 nm/ns (blue), and 0.1 nm/ns (green, 0.4-ns running average in light green). (**B**) Snapshots of unbinding trajectory during stretching simulation at 0.1 nm/ns (Sim3d; S7 Table). Springs indicate position and direction of applied forces. Top panel shows complex at the beginning of the simulation; other panels show snapshots at 3 time points indicated with black arrow heads in (A). End-to-end distance indicates the distance between the stretched atoms on the C-termini of EC2. (**C**) Force versus end-to-end distance for simulations of the largest crystallographic *trans* interface (S10A Fig) observed for the *hs* PCDH24 EC1-2 II structure (simulations Sim4b-Sim4d; S7 Table, S10 Data). Forced unbinding simulations were carried out at stretching speeds of 10 nm/ns (red), 1 nm/ns (blue), and 0.1 nm/ns (green, 2-ns running average in light green). PCDH24, protocadherin-24.
(PDF)

**S10 Fig. Crystal contacts in the *hs* PCDH24 EC1-2 II structure.** (**A**) Ribbon diagram of 2 monomers showing a crystal contact between chains C and D with an interface area of 1,019.3 Å$^2$. The arrangement is antiparallel, likely describing a possible *trans* interface. N-glycosylation at p.N9 is shown in orange licorice. A similar arrangement is seen in chains A and B with an interface area of 1,022.8 Å$^2$. (**B**) A crystal contact between chains B and D (435.5 Å$^2$) with the F-G loops of EC2 on top of each other (black box). (**C**) The dimer shown in (A) forms a dimer of dimers with chains A and B to form the asymmetric unit and includes 4 additional interfaces with areas of 215.1 Å$^2$ (chains A and D), 417.6 Å$^2$ (chains A and C), 435.5 Å$^2$ (chains D and B, shown in (B)), and 278.5 Å$^2$ (chains B and C). (**D**) An additional *trans* overlap exists between chains C and D (353.1 Å$^2$). (**E, F**) Two smaller interfaces between chain D and B (344.7 Å$^2$) and chains A and D (282.1 Å$^2$) are shown in (E) and (F), respectively. (**G**) Hypothetical heterotetrameric junction observed in the crystal's asymmetric unit. PCDH24, protocadherin-24.
(PDF)

**S11 Fig. Crystal contacts in the *mm* PCDH24 EC1-3 structure.** (**A**) Crystal contacts show a potential *cis* trimer formed by 3 parallel monomers. The interface area between 2 monomers is 1,352.6 Å$^2$. Two monomers are shown as ribbons, while the third one is shown in molecular surface representation. (**B**) Detail of trimer seen from EC1 (top) shows that the extended N-terminal strands interlace between monomers. (**C**) Crystal contacts show an antiparallel *trans* dimer with an interface area of 1,221.2 Å$^2$. (**D**) Taken together, the potential *cis* and *trans* interfaces form a large complex with 2 *cis* trimers forming an antiparallel *trans* dimer.

PCDH24, protocadherin-24.
(PDF)

**S12 Fig. Sequence conservation of PCDH24 EC1-3.** (**A**) Surface representation of *mm* PCDH24 EC1-3 structure with residues colored according to sequence conservation determined using Consurf and a sequence alignment including over 94 species (S5 Table). Teal colors indicate residues that are least conserved, while magenta indicates residues that are most conserved among species. (**B**) Transparent surface representation of *mm* PCDH24 EC1-3 with most conserved residues shown as an opaque magenta surface. Protein core is conserved. PCDH24, protocadherin-24.
(PDF)

**S13 Fig. Homophilic binding assays of *hs* PCDH24 at various time points.** (**A-H**) Protein G beads coated with the Fc-tagged full-length extracellular domain of *hs* PCDH24 (A) and its C-terminal truncation versions (B-H). Images show bead aggregation observed at the start of the experiment (T0), after 30 min (T30), after 60 min (T60) followed by rocking for 1 min (R1), all in the presence of 2 mM CaCl$_2$. Bar– 500 μm. (**I**) Protein G beads coated with the Fc-tagged full-length extracellular domain of *hs* PCDH24 in the presence of 2 mM EDTA, shown as in (A). PCDH24, protocadherin-24.
(PDF)

**S14 Fig. Homophilic binding assays of *mm* CDHR5 at various time points.** (**A-D**) Protein G beads coated with the Fc-tagged full-length cadherin extracellular domain of *mm* CDHR5 (A) and its C-terminal truncation versions (B-D). Images show the aggregation observed at the start of experiment (T0), after 60 min (T60) followed by rocking for 1 min (R1) and 2 min (R2) in the presence of 2 mM CaCl$_2$. Bar– 500 μm. (**E**) Protein G beads coated with the Fc-tagged full-length cadherin extracellular domain of *mm* CDHR5 in the presence of 2 mM EDTA shown as in (A). CDHR5, cadherin-related family member 5.
(PDF)

**S15 Fig. Sequence alignment of *hs* and *mm* CDHR5 EC repeats.** All 4 EC repeats for each species are aligned to each other (EC1 to EC4). CDHR5, cadherin-related family member 5; EC, extracellular cadherin.
(PDF)

**S16 Fig. Crystal contacts in the *hs* CDHR5 EC1-2 structure.** (**A-F**) Ribbon diagrams showing various crystal contacts between monomers of *hs* CDHR5 EC1-2. Interface areas are 1,433.9 Å$^2$ (A), 530.2 Å$^2$ (B), 364.7 Å$^2$ (C), 347.9 Å$^2$ (D), 132.7 Å$^2$ (E), and 114.0 Å$^2$ (F), respectively. Interfaces in (A) and (B) correspond to possible *trans* overlaps of EC1-2 and EC1-3, respectively. While *hs* CDHR5 does not mediate *trans* homophilic adhesion based on the homophilic binding assay data presented in Fig 4, these interfaces may serve as templates for the mouse CDHR5 protein. Crystal contacts in (D-F) are unlikely to be of physiological relevance due to the arrangement of the monomers. CDHR5, cadherin-related family member 5.
(PDF)

**S17 Fig. Sequence conservation of CDHR5 EC1-2.** (**A**) Surface representation of *hs* CDHR5 EC1-2 structure with residues colored according to sequence conservation determined using Consurf and a sequence alignment including over 69 species (S6 Table). Teal colors indicate residues that are least conserved, while magenta indicates residues that are most conserved among species. (**B**) Transparent surface representation of *hs* CDHR5 EC1-2 with most conserved residues shown as an opaque magenta surface. CDHR5, cadherin-

related family member 5.
(PDF)

**S18 Fig. Heterophilic binding assays of *hs* and *mm* PCDH24 and CDHR5.** (**A, B**) Images from binding assays of Fc-tagged *hs* PCDH24 (full-length extracellular domain and EC1-4Fc) mixed with the Fc-tagged full-length *hs* CDHR5 cadherin extracellular domain (A) and Fc-tagged *mm* PCDH24 (full-length extracellular domain and EC1-4Fc) mixed with Fc-tagged full-length *mm* CDHR5 cadherin extracellular domain (B). Green fluorescent Protein A beads are coated with Fc-tagged PCDH24 fragments and red fluorescent Protein A beads are coated with Fc-tagged CDHR5 in all panels. The white boxes indicate the regions shown in Fig 7. Images show bead aggregation observed after 60 min followed by rocking for 2 min in the presence of 2 mM CaCl$_2$. (**C-F**) Images from binding assays of truncations of Fc-tagged *hs* CDHR5 mixed with *hs* PCDH24 EC1-4Fc (C), truncations of Fc-tagged *hs* PCDH24 mixed with *hs* CDHR5 EC1-4Fc (D), truncations of Fc-tagged *mm* CDHR5 mixed with *mm* PCDH24 EC1-4Fc (E), and truncations of Fc-tagged *mm* PCDH24 mixed with *mm* CDHR5 EC1-4Fc (F). Images show bead aggregation observed after 60 min followed by rocking for 2 min in the presence of 2 mM CaCl$_2$. (**G**) Images from binding assays of minimum EC repeats required for heterophilic adhesion of *hs* and *mm* PCDH24 and CDHR5. The minimum units for heterophilic adhesion for the human proteins are CDHR5 EC1Fc and PCDH24 EC1-2Fc. The minimum units for heterophilic adhesion for the mouse proteins are CDHR5 EC1-2Fc and PCDH24 EC1-2Fc. Images show bead aggregation observed after 60 min followed by rocking for 2 min in the presence of 2 mM CaCl$_2$. (**H**) Protein A beads coated with Fc-tagged full-length *hs* and *mm* PCDH24 and CDHR5 cadherin extracellular domains (including MAD10 for PCDH24 when indicated and without the MLD for CDHR5) in the presence of 2 mM EDTA shown as in (A, B). Bar– 500 μm. (**I**) Aggregate size (S11 Data) for minimum units of heterophilic adhesion of Fc-tagged *hs* and *mm* PCDH24 and CDHR5 at the start of the experiment (T0), after 60 min (T60) followed by rocking for 1 min (R1) and 2 min (R2). (**J**) Aggregate size (S12 Data) for full-length *hs* and *mm* PCDH24 and CDHR5 at the start of the experiment (T0), after 60 min (T60) followed by rocking for 1 min (R1) and 2 min (R2). Error bars in I and J are standard error of the mean (*n* indicated in S4 Table). CDHR5, cadherin-related family member 5; EC, extracellular cadherin; MLD, mucin-like domain; PCDH24, protocadherin-24.
(PDF)

**S19 Fig. SPR experiments testing the minimal adhesive unit for mouse PCDH24 and CDHR5.** Raw data from the SPR experiment performed on *mm* PCDH24 EC1-2Fc and *mm* CDHR5 EC1-2 shown in green gradient from lowest concentration (light green) to highest concentration (dark green) of *mm* CDHR5 EC1-2, with fits in black. The average affinity $K_D$ and rates ($k_{on}$ and $k_{off}$) are indicated (*n* = 1). Values were obtained from a kinetic analysis of the raw data (S13 Data). CDHR5, cadherin-related family member 5; PCDH24, protocadherin-24; SPR, surface plasmon resonance.
(PDF)

**S20 Fig. Potential PCDH24 EC1-2/CDHR5 EC1-2 complex interfaces.** Complexes were built by aligning protein fragments to monomers forming the largest antiparallel *trans* crystallographic interface in the *hs* PCDH24 EC1-2 II structure. (**A**) Molecular surface representation of *hs* PCDH24 EC1-2 (light blue) and *hs* CDHR5 EC1-2 (dark green) with a potential interface surface (silver) exposed. Interfacing residues are listed. Orange labels highlight residues that remained buried at the interface during a 10-ns long equilibrium simulation. Residues in *hs* CDHR5 EC2 are in gray as the minimum unit for heterophilic adhesion includes *hs* CDHR5 EC1. (**B**) Molecular surface representation of *mm* PCDH24 EC1-2 (dark blue) and *mm*

CDHR5 EC1-2 (light green) with a potential interface surface (silver) exposed. Interfacing residues are listed. CDHR5, cadherin-related family member 5; PCDH24, protocadherin-24.
(PDF)

**S1 Table. Percent identity of EC repeats for CDH23, PCDH15, PCDH24, and CDHR5 proteins across species.** The accession numbers of the species used are listed in S2 Table.
(PDF)

**S2 Table. Accession numbers of CDHR5, PCDH24, PCDH15, and CDH23 sequences used for alignment of EC repeats across different species.**
(PDF)

**S3 Table. Accession numbers of PCDH24, CDHR5, CDH23, PCDH15, CDH1, and CDH2 sequences used for generating Fig 1B.**
(PDF)

**S4 Table. Number of biological replicates for bead aggregation assays.**
(PDF)

**S5 Table. Accession numbers of PCDH24 sequences used in Consurf.**
(PDF)

**S6 Table. Accession numbers of CDHR5 sequences used in Consurf.**
(PDF)

**S7 Table. Summary of simulations.**
(PDF)

**S1 Data. Source file of bead aggregate size for *hs* PCDH24 homophilic binding.** This Excel file contains the source data for bead aggregate sizes in Fig 3L. The mean of means, standard deviation, and standard errors for the constructs are listed in the Excel file.
(XLSX)

**S2 Data. Source file of bead aggregate size for *hs* PCDH24 EC1-2 and EC1-3 mutant homophilic binding.** This Excel file contains the source data for the bead aggregate sizes in Fig 4B. The mean of means, standard deviation, and standard errors for the constructs are listed in the Excel file.
(XLSX)

**S3 Data. Source file of bead aggregate size for *hs* and *mm* PCDH24 EC1-10 homophilic binding.** This Excel file contains the source data for the bead aggregate sizes in Fig 5E. The mean of means, standard deviation, and standard errors for the constructs are listed in the Excel file.
(XLSX)

**S4 Data. Source file of bead aggregate size for *hs* and *mm* CDHR5 EC1-4 homophilic binding.** This Excel file contains the source data for the bead aggregate sizes in Fig 5F. The raw aggregate area, mean of means, standard deviation, and standard errors for the constructs are listed in the Excel file.
(XLSX)

**S5 Data. Source file of bead aggregate size for *mm* CDHR5 homophilic binding.** This Excel file contains the source data for the bead aggregate sizes in Fig 6H. The raw aggregate area, mean of means, standard deviation, and standard errors for the constructs are listed in the Excel file.
(XLSX)

**S6 Data. Source file of bead aggregate size for *hs* PCDH24 and CDHR5 heterophilic binding.** This Excel file contains the source data for the bead aggregate sizes in Fig 7I. The raw aggregate area, mean of means, standard deviation, and standard errors for the constructs are listed in the Excel file.
(XLSX)

**S7 Data. Source file of bead aggregate size for *mm* PCDH24 and CDHR5 heterophilic binding.** This Excel file contains the source data for the bead aggregate sizes in Fig 7J. The raw aggregate area, mean of means, standard deviation, and standard errors for the constructs are listed in the Excel file.
(XLSX)

**S8 Data. Source file of force versus end-to-end distance in simulations of the *mm* PCDH24 EC1-3 monomer.** This text file contains the source data for forces and end-to-end distances in S6D Fig.
(ZIP)

**S9 Data. Source file of force versus end-to-end distance in simulations of the *mm* PCDH EC1-2 I complex.** This text file contains the source data for forces and end-to-end distances in S9A Fig.
(ZIP)

**S10 Data. Source file of force versus end-to-end distance in simulations of the *mm* PCDH EC1-2 II complex.** This text file contains the source data for forces and end-to-end distances in S9C Fig.
(ZIP)

**S11 Data. Source file of bead aggregate size for *hs* and *mm* PCDH24 and CDHR5 heterophilic binding (minimum adhesive units).** This Excel file contains the source data for the bead aggregate sizes in S18I Fig. The raw aggregate area, mean of means, standard deviation, and standard errors for the constructs are listed in the Excel file.
(XLSX)

**S12 Data. Source file of bead aggregate size for *hs* and *mm* PCDH24 and CDHR5 heterophilic binding (EDTA controls).** This Excel file contains the source data for the bead aggregate sizes in S18J Fig. The raw aggregate area, mean of means, standard deviation, and standard errors for the constructs are listed in the Excel file.
(XLSX)

**S13 Data. Source file for SPR experimental results.** This Excel file contains the source data associated with the SPR data shown in S19 Fig. The response data at normal resolution for channel 1 and channel 2 are included, as well as the corrected data, which were plotted. Buffer blanks are included following their respective analyte injections as well.
(XLSX)

**S1 Raw Images. Original gel images for western blots.** This pdf file contains original gel images used to generate Figs 3M, 4J, 5G, 6I and 7K.
(PDF)

## Acknowledgments

We thank members of the Sotomayor laboratory and Harper Smith for assistance and discussions. Use of APS NE-CAT beamlines was supported by NIH (P41 GM103403 and S10

RR029205) and the Department of Energy (DE-AC02-06CH11357) through grants GUP 49774, 59251, and 70086. Use of TACC-Stampede and PSC-Bridges supercomputers was supported by the National Science Foundation through XSEDE (XRAC MCB140226). Use of the OSC-Owens supercomputer was supported by grants PAS1037 and PAA0217. D.M. was a Pelotonia fellow, and M.S. was an Alfred P. Sloan fellow (FR-2015-65794).

## Author Contributions

**Conceptualization:** Michelle E. Gray, Zachary R. Johnson, Debadrita Modak, Matthew J. Tyska, Marcos Sotomayor.

**Data curation:** Michelle E. Gray, Zachary R. Johnson, Debadrita Modak, Marcos Sotomayor.

**Formal analysis:** Michelle E. Gray, Zachary R. Johnson, Debadrita Modak, Elakkiya Tamilselvan, Marcos Sotomayor.

**Funding acquisition:** Matthew J. Tyska, Marcos Sotomayor.

**Investigation:** Michelle E. Gray, Zachary R. Johnson, Debadrita Modak, Elakkiya Tamilselvan, Marcos Sotomayor.

**Methodology:** Michelle E. Gray, Zachary R. Johnson, Debadrita Modak, Marcos Sotomayor.

**Project administration:** Matthew J. Tyska, Marcos Sotomayor.

**Resources:** Matthew J. Tyska, Marcos Sotomayor.

**Supervision:** Matthew J. Tyska, Marcos Sotomayor.

**Writing – original draft:** Michelle E. Gray, Debadrita Modak, Marcos Sotomayor.

**Writing – review & editing:** Michelle E. Gray, Zachary R. Johnson, Debadrita Modak, Elakkiya Tamilselvan, Matthew J. Tyska, Marcos Sotomayor.

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
