## [Editor Report · Decision Letter 0]

22 Sep 2020

Dear Dr Sotomayor, 

Thank you for submitting your manuscript entitled "Species-Dependent Heterophilic and Homophilic Cadherin Interactions in Intestinal Intermicrovillar Links" for consideration as a Research Article by PLOS Biology.

Your manuscript has now been evaluated by the PLOS Biology editorial staff as well as by an academic editor with relevant expertise and I am writing to let you know that we would like to send your submission out for external peer review.

Please re-submit your manuscript within two working days, i.e. by Sep 24 2020 11:59PM.

Kind regards,

Ines

--

Ines Alvarez-Garcia, PhD

Senior Editor

PLOS Biology

---

## [Decision Letter · Decision Letter 1]

14 Dec 2020

Dear Dr Sotomayor,

Thank you very much for submitting your manuscript "Species-Dependent Heterophilic and Homophilic Cadherin Interactions in Intestinal Intermicrovillar Links" for consideration as a Research Article at PLOS Biology. Thank you also for your patience as we completed our editorial process, and please accept my sincere apologies for the long delay in providing you with our decision. Your manuscript has been evaluated by the PLOS Biology editors, an Academic Editor with relevant expertise, and by three independent reviewers.

As you will see, the reviewers think your findings are interesting and significant for the field, however they also raise several concerns that need to be addressed. These include experimental confirmation of the molecular interactions proposed to validate the adhesive models, to perform mutagenesis studies to test the putative biological interfaces suggested by crystal packing interactions that are buried among surface areas, and to address experimentally several questions.

In light of the reviews (attached below), we will not be able to accept the current version of the manuscript, but we would welcome re-submission of a much-revised version that takes into account the reviewers' comments. We cannot make any decision about publication until we have seen the revised manuscript and your response to the reviewers' comments. Your revised manuscript is also likely to be sent for further evaluation by the reviewers.

We expect to receive your revised manuscript within 3 months. 

**IMPORTANT - SUBMITTING YOUR REVISION**

*Re-submission Checklist*

*Published Peer Review*

*PLOS Data Policy*

*Blot and Gel Data Policy*

Sincerely,

Ines

--

Ines Alvarez-Garcia, PhD,

Senior Editor,

PLOS Biology

Reviewers’ comments

Rev. 1:

In their study, Gray et al. present a crystal structure of the extracellular cadherin repeats (EC) 1-2 of human protocadherin-24 (EC1-3 for the mouse protein) and EC domain 1-2 of the human Protocadherin Cadherin-related family member 5 (CDHR5) at very high resolution (2.3Å, 2.1Å and 1.9Å respectively). The heterophilic interaction between these two protocadherins has previously been reported to be essentially for microvilli development and function. Notably, the authors reveal a rather low sequence conservation of both proteins (in contrast to similar adhesion molecules of the classical cadherin and protocadherin family) and important structural differences between mouse and human proteins.

As a functional readout, the authors then used aggregation assays to demonstrate that human PCDH24 forms calcium-dependent self-aggregates (in trans) when equipped with a minimum of three EC repeats. In line with the structural differences described in this study, the extracellular domain of mouse PCDH24 does not aggregate. Notably, CDHR5 behaves the other way around: Mouse CDHR5 self-aggregates, whereas PCDH24 does not. Testing the important heterophilic aggregation between PCDH24 and CDHR5, the authors reveal that proteins of both species are capable of forming calcium-dependent hetero-aggregates.

The manuscript is well written and addresses an issue which is of high relevance for our understanding of microvilli development and function. The presented data are robust and well described. However, some important issues remain unclear and should be addressed to provide a complete picture of the molecular mechanism of the PCDH24-CDH5 module.

Major points:

1. The calculation of the average identity across species should be better explained in the text and in the figure legend.

2. Page 17, line 9-10 (and discussion): The authors cannot conclude that PCDH24 EC1 exhibits a calcium-independent aggregation, because they argue some lines above that the calcium-independent aggregation of EC1 might be explained by a stabilization of the Fc Tag. This discrepancy should be tested in a different experimental setup.

3. Some important questions remain and should be addressed experimentally:

a. Is homophilic aggregation of human PCDH24 (and mouse CDHR5) a prerequisite for the formation of hetero-aggregates?

b. Does cis-homo-dimerization occur in case of human PCDH24 and/or mouse CDHR5?

c. Is the (cis or trans) homo-dimerization of one of the adhesion partners essential for trans-heterophilic aggregation?

d. Is trans-heterophilic aggregation stronger than trans-homophilic interaction/are there binding preferences for either homo- or heterophilic interactions?

Minor points:

Introduction: page 10, line 10. Some diseases should be given as examples (with more recent citations)

Page 10, line 15: The abbreviation CDHR should be introduced at first use

Page 12, line 5: PCDH24, CDHR5 and PCDH15 are not (classical) "cadherins"

Rev. 2:

This is a well-performed study that combines X-ray crystallography and bead aggregation assays to present the structural basis for the species dependent adhesive properties of PCDH24 and CDHR5, two non-classical cadherins that function in maintaining the intermicrovillar links in the small intestine. The authors determined the crystal structures of select N-terminal fragments from PCDH24 and CDHR5. By combining the structural observations with bead aggregation analysis, the authors report that homophilic and heterophilic interactions by PCDH24 and CDHR5 are species dependent involving distinct minimal adhesive units in different species. Overall, this work is of high quality and the results are significant. The data presented in this work will be of interest to readers studying cell adhesion molecules and in particular those interested in the molecular diversity of cadherins.

My main criticisms are:

1. Lack of solution biophysical characterization to validate the proposed adhesive models. Bead aggregation is a reasonably good proxy to cell aggregation behavior, but rigorous solution biophysical methods such as analytical ultracentrifugation or isothermal titration calorimetry will be needed to corroborate the molecular interaction mechanisms proposed.

2. Lack of mutagenesis studies to test the putative biological interfaces suggested by crystal packing interactions that bury meaningful amount of surface areas.

Some specific comments:

1. Page 5, 2nd paragraph and Figure 1: "average sequence identity is 51.6% for CDH23 EC repeats and 45.5% for PCDH15 repeats……In contrast, average percent identity is 16.4% for PCDH24 and 11% for CDHR5, considerably lower when compared to values for CDH23 and PCDH15." I gather that the main point the authors want to make here is that the sequence identities for CDH23 and PCDH15 are higher than those for PCDH24 and CDHR5. But to make meaningfully quantitative comparisons between these two pairs using these numbers (with three or four significant digits), the exact same set of species (preferably from as diverse phyla as possible), needs to be selected for the alignment of each cadherin. But from Tables S2-S5, it seems that the set of species selected is not completely identical for each of these four cadherins. For these comparisons, it would be better to report a range of numbers rather than specific average numbers.

2. Page 7, line 7: (S187:S201) needs to be changed to (C187:C201).

3. Page 7, lines 23-24: "equilibrium and steered molecular dynamics (SMD) simulations of mm PCDH24 EC1-3 show that rigidity and mechanical strength are not compromised in this non-canonical linker." The simulations suggested that this linker is rigid, but what are the possible implications of this rigidity? This claim needs to be discussed further in detail or removed from the manuscript if it has little to do with the adhesive interaction mechanisms proposed in this work.

4. Page 8, 1st paragraph: "The sequence motifs involved in calcium coordination at site 0 are conserved, suggesting that crystallization conditions, including 3M LiCl, facilitated the opening of this site." Possible, but not certain without further experimental evidence, so I would use "may have facilitated" or something like that. Lithium chloride may or may not have facilitated the opening of this particular site. But if, as stated in this sentence, 3M lithium chloride in the crystallization solution disrupted calcium binding at site 0, which is clearly not a physiologically relevant condition, no functional implications can be deduced from the structures observed under this condition. Thus, it raises a question about the proposed cis interaction mechanism for mouse PCDH24 by N-terminal interlocking.

5. Page 8, the section of crystal contacts: as mentioned above in my main criticisms, a number of crystal packing interactions with reasonable buried surface areas were discussed in detail, but all just discussions with no experimental validations. Testing mutants designed to disrupt these interfaces would make this a much stronger paper.

6. Page 9, lines 13-14: why use (p.N161) and (p.N281) instead of just (N161) and (N281) when referring to these glycosylation sites? The same comment applies to page 12, lines 6-7 where glycosylation sites are mentioned.

Rev. 3:

This manuscript by Gray et al., tries to identify the structural determinants of the interactions between protocadherin-24 (PCDH24) and CDHR5. PCDH24 and CDHR5 are non-classical cadherins and their interactions drive the assembly of the intestinal brush border assembly. The authors present two x-ray crystal structures of PCDH24 (one from mouse and anther human) in addition to a human CDHR5 structure, all in their monomeric forms. These structures confirm similarities between PCDH24 to CDH23 and between CADHR5 and PCDH15. Several crystal contacts are described and it is suggested that these may facilitate the adhesion interfaces. In addition, the authors present bead aggregation assay to characterize the protein fragments that are required for adhesion. Using the aggregation assays with both the human and mouse proteins the authors suggest that these proteins exhibit species-dependent homophilic interactions and species-distinct heterophilic adhesive units. The technical aspects in this paper are well executed and are described in sufficient details.

My major criticism lies in the fact that it does not appear that the findings and conclusions of this current manuscript contribute to major advances in this field. In the absence of structures that include the adhesive interfaces, the physiological relevance of the crystal contacts presented here remain unclear. This criticism holds true for both heterophilic and homophilic interactions. Ultimately the reader does not obtain an answer regarding the structural determinants of the interactions between PCDH24 and CADH24. In the absence of structure that describe the interface I cannot recommend to accept this paper to Plos biology.

---

## [Decision Letter · Decision Letter 2]

18 Aug 2021

Dear Dr Sotomayor,

Thank you for submitting your revised Research Article entitled "Species-Dependent Heterophilic and Homophilic Cadherin Interactions in Intestinal Intermicrovillar Links" for publication in PLOS Biology. I have now obtained advice from the three original reviewers and have discussed their comments with the Academic Editor. 

The reviews are attached below. Based on the reviews, we will probably accept this manuscript for publication, provided you satisfactorily address the policy-related requests stated below.

In addition, we would like to suggest a change in the titled to improve it:

"Heterophilic and homophilic cadherin interactions in intestinal intermicrovillar links are species-dependent"

We expect to receive your revised manuscript within two weeks. 

*Published Peer Review History*

*Early Version*

Sincerely,

Ines

--

Ines Alvarez-Garcia, PhD,

Senior Editor,

ialvarez-garcia@plos.org,

PLOS Biology

Fig. 1A; Fig. 3L; Fig. 4B; Fig. 5E, F; Fig. 6H; Fig. 7I, J; Fig. S6D; Fig. S9A, C; Fig. S18I, J and Fig. S19

** Also, we have noted that the data deposited in the Protein data bank with accession number 7N86 is not publicly available, so please make it available now. In addition, if the raw images for bead aggregation assays are relevant for the manuscript, you should deposit them in a public database.

We require the original, uncropped and minimally adjusted images supporting all blot and gel results reported in an article's figures or Supporting Information files. We will require these files before a manuscript can be accepted so please prepare and upload them now. Please carefully read our guidelines for how to prepare and upload this data: https://journals.plos.org/plosbiology/s/figures#loc-blot-and-gel-reporting-requirements 

DATA NOT SHOWN?

Reviewers’ comments

Rev. 1:

The authors fully addressed all of my concerns and I recommend the manuscript for publication in PLOS Biology.

Rev. 2:

The authors addressed all my concerns and comments in a satisfactory way. This revised manuscript is clearly presented and makes a strong case that homophilic and heterophilic interactions by PCDH24 and CDHR5 are species dependent involving distinct minimal adhesive units in different species.

Rev. 3:

The authors have put forth a thorough and scholarly effort to respond to essentially all of the reviewer's comments. The addition of a second structure of hs PCDH24 EC1-2 that potentially shows a trans interface and the additional binding assays significantly strengthen the paper. I think that the article, in its current format, warrants publication in PLOS Biology.

---

## [Editor Report · Decision Letter 3]

30 Oct 2021

Dear Dr Sotomayor,

On behalf of my colleagues and the Academic Editor, Kylie Walters, I am pleased to say that we can in principle accept your Research Article "Heterophilic and Homophilic Cadherin Interactions in Intestinal Intermicrovillar Links are Species Dependent" for publication in PLOS Biology, provided you address any remaining formatting and reporting issues. These will be detailed in an email that will follow this letter and that you will usually receive within 2-3 business days, during which time no action is required from you. Please note that we will not be able to formally accept your manuscript and schedule it for publication until you have any requested changes.

PRESS

Sincerely, 

Ines

--

Ines Alvarez-Garcia, PhD 

Senior Editor 

PLOS Biology
